# Mercury mobility, colloid formation and methylation in a polluted fluvisol as affected by manure application and flooding-draining cycle.

Lorenz Gfeller[1], Andrea Weber[1], Isabelle Worms[2], Vera I. Slaveykova[2], Adrien Mestrot[1]

[1]Institute of Geography, University of Bern, Hallerstrasse 12, 3012 Bern, Switzerland
[2]Environmental Biogeochemistry and Ecotoxicology, Department F.-A. Forel for environmental and aquatic sciences, School of Earth and Environmental Sciences, Faculty of Sciences, University of Geneva, Uni Carl Vogt, Bvd Carl-Vogt 66, CH-1211 Geneva 4, Switzerland
*Correspondence to*: Adrien Mestrot (adrien.mestrot@giub.unibe.ch)

**Abstract**

Floodplain soils polluted with high levels of mercury (Hg) are potential point sources to downstream ecosystems. Repeated flooding (e.g. redox cycling) and agricultural activities (e.g. organic matter addition) may influence the fate and speciation of Hg in these soil systems. The formation and aggregation of colloids and particles influences both Hg mobility and its bioavailability to methylmercury (MeHg) forming microbes. In this study, we conducted a microcosm flooding-draining experiment on Hg polluted floodplain soils originating from an agriculturally used area situated in the Rhone Valley (Valais, Switzerland). The experiment comprised two 14 days flooding periods separated by one 14 days draining period. The effect of freshly added natural organic matter on Hg dynamics was assessed by adding liquid cow manure (+MNR) to two soils characterized by different Hg ($47.3 \pm 0.5$ mg kg$^{-1}$ or $2.38 \pm 0.01$ mg kg$^{-1}$) and organic carbon (OC: 1.92 wt. % or 3.45 wt. %) contents. During the experiment, the release, colloid formation of Hg in soil solution and the net MeHg production in the soil were monitored. Upon manure addition in the highly polluted soil (lower OC), an accelerated release of Hg to the soil solution could be linked to a fast reductive dissolution of Mn oxides. The manure treatments showed a fast sequestration of Hg and a higher percentage of particulate ($0.02 – 10$ µm) bound Hg. As well, analyses of soil solutions by asymmetrical flow field-flow fractionation coupled with inductively coupled plasma mass spectrometry (AF4–ICP–MS) revealed a relative increase of colloidal Hg bound to dissolved organic matter (Hg-DOM) and inorganic colloidal Hg (70 - 100 %) upon manure addition. Our experiment shows a net MeHg production the first flooding and draining period and a subsequent decrease in absolute MeHg concentrations after the second flooding period. Manure addition did not change net MeHg production significantly in the incubated soils. The results of this study suggest that manure addition may promote Hg sequestration by Hg complexation on large organic matter components and the formation and aggregation of inorganic HgS$_{(s)}$ colloids in Hg polluted fluvisols with low levels of natural organic matter.

**1. Introduction**
Mercury (Hg) is a pollutant of global concern due to its high toxicity and to its global biogeochemical cycle which
spans all environmental compartments (atmosphere, oceans, soils etc.) (Beckers and Rinklebe, 2017; AMAP/UN
Environment, 2019). Sediments and soils are major Hg pools with relatively long residence times (Amos et al., 2013;
Driscoll et al., 2013). Legacy Hg from industrial sites (e.g. chloralkali plants or mining areas) retained in soils are a
key source for present day atmospheric Hg (Amos et al., 2013). However, this retained Hg pool can also be remobi-
lized by landscape alteration, land use (e.g. fertilization, manure addition) or climate induced changes such as drought-
flood-drought cycles of soils (Singer et al., 2016). These inputs are a threat to downstream ecosystems and human
health due to release of inorganic Hg and the formation and bioaccumulation of toxic monomethylmercury (MeHg)
in both aquatic and terrestrial food chains (Bigham et al., 2017).
Mercury is redox sensitive and occurs mainly as elemental $Hg^0$, inorganic $Hg^{2+}$ or in the form of MeHg in soils. In
general, Hg speciation in soils depends on the biogeochemical conditions. For example, in natural organic matter
(NOM) rich boreal peatlands and forest soils, Hg is primarily bound to thiol-groups of NOM (NOM–Hg), associated
with $FeS_{(s)}$ or found as cinnabar ($HgS_{(s)}$) or meta-cinnabar ($ß\text{-}HgS_{(s)}$).These species are the thermodynamically most
favored forms of Hg in these environments (Skyllberg et al., 2006; Skyllberg and Drott, 2010; Biester et al., 2002).
However, Hg sorbed on the surfaces of manganese (Mn), iron (Fe) and aluminum (Al) oxy-hydroxides may also
represent important Hg-pools in soils with low amounts of NOM (Guedron et al., 2009).
The fate of Hg in soils is still not well characterized, and its mobilization and sequestration in soil depends on a variety
of factors and mechanisms. The release of Hg to the soil solution and its further transport has been associated with the
mobilization of NOM (Kronberg et al., 2016; Eklöf et al., 2018; Åkerblom et al., 2008), copper (Cu) nanoparticles
(Hofacker et al., 2013) or the reductive dissolution of Fe/Mn-oxyhydroxides (Frohne et al., 2012; Gygax et al., 2019;
Poulin et al., 2016). Earlier studies reported a relatively rapid decrease of dissolved Hg after its release upon flooding
in various riparian settings (Hofacker et al., 2013; Poulin et al., 2016; Gygax et al., 2019). Possible pathways for this
decrease are $Hg^{2+}$ reduction to $Hg^0$, sorption to recalcitrant NOM, formation of meta-cinnabar $ß\text{-}HgS_{(s)}$ or co-precipi-
tation of Hg in sulphides (e.g. $FeS_{(s)}$) or metallic particles.
Metallic colloids in soil may be formed by biomineralization during soil reduction or precipitation in the root zone
and potentially incorporate toxic trace elements like Hg (Weber et al., 2009; Manceau et al., 2008). These colloids
may increase the mobility and persistence of toxic trace metals in soil solution if they do not aggregate to bigger
particles. During a flooding incubation experiment, Hofacker et al. (2013) observed the incorporation of Hg in Cu
nano-particles, which were shown to be formed by fermetive bacteria species (Hofacker et al., 2015). Colloidal ß-
$HgS_{(s)}$ has been reported to form abiotically in soils under oxic conditions directly by interaction with thiol-groups of
NOM (Manceau et al., 2015). In solution, Dissolved Organic Matter (DOM) has a major influence in the formation
and aggregation of metallic colloids and particles. It may promote the dissolution of $HgS_{(s)}$ phases, decelerate the
aggregation and growth of $HgS_{(s)}$ colloids as well as affect the crystallinity of $HgS_{(s)}$ phases (Miller et al., 2007;
Ravichandran et al., 1998; Gerbig et al., 2011; Poulin et al., 2017; Pham et al., 2014). Same effects were also observed
for other metal sulphide-, oxide- or carbonate colloids (Aiken et al., 2011; Deonarine et al., 2011). In case of Hg,
inhibition of ß-$HgS_{(s)}$ formation may in turn increase its mobility and bioavailability to MeHg producing microorgan-
isms (Deonarine and Hsu-Kim, 2009; Ravichandran et al., 1999; Aiken et al., 2011; Graham et al., 2012). Chelation
of Hg with higher molecular weight NOM may as well inhibit the microbial availability of Hg (Bravo et al., 2017).
Within Hg–NOM, hydrophobic, thiol rich NOM with higher molecular weight contain a higher density of strong
sorption sites (thiol groups) (Haitzer et al., 2002). However, different ligand exchange reactions (e.g. carboxyl-groups
to thiol groups) kinetically control this sorption and thus the bioavailability of dissolved Hg in aqueous systems (Miller
et al., 2007; Miller et al., 2009; Liang et al., 2019). The partly contradicting statements above illustrate the complex
role of NOM and DOM on the Hg cycle and Hg bioavailability and the need for more research in this field.
The formation of MeHg from inorganic $Hg^{2+}$ has been shown to be primarily microbially driven. Environments of
redox oscillation (e.g. floodplains, estuaries) represent hot spots for Hg methylation (Marvin-DiPasquale et al., 2014;
Bigham et al., 2017). Mercury methylators are usually anaerobe microbial species such as sulphate reducers (SRB),
Fe reducers (FeRB), archaea and some firmicutes (Gilmour et al., 2013). Generally, Hg is bioavailable to methylators
in the form of dissolved $Hg^{2+}$, Hg complexed by labile DOM, Hg bearing inorganic nanoparticles (e.g. $FeS_{(s)}$, $HgS_{(s)}$)
but is less available when complexed by particulate organic matter (Hg–POM) or larger inorganic particles (Chiasson-
Gould et al., 2014; Graham et al., 2013; Rivera et al., 2019; Zhang et al., 2012; Jonsson et al., 2012). Further, DOM
is a main driver of Hg methylation as it influences both bioavailability and microbial activity. The role of DOM as
electron donor may enhance the microbial activity and thus the cellular uptake. The composition and origin of DOM
were reported to change Hg methylation rates (Drott et al., 2007; Bravo et al., 2017). For example, Bravo et al. (2017)
showed that in lake sediments, terrestrial derived DOM led to slower methylation rates than phytoplankton derived
DOM. The addition of DOM in form of organic amendments (e.g. manure, rice straw, biochar) has been reported to
have both an enhancing (Gygax et al., 2019; Liu et al., 2016; Wang et al., 2019; Eckley et al., 2021; Wang et al., 2020)
or no effect (Zhu et al., 2016; Liu et al., 2016) on the net MeHg production in soils. Further, organic amendments
were reported to shift microbial communities. Both the enhancement of Hg demethylators, Hg reducers (Hu et al.,
2019) as well as the enhancement Hg methylators upon organic amendments were reported (Tang et al., 2019; Wang
et al., 2020). Environments of  elevated Hg methylation (riparian zone, estuary) are  also  places of elevated NOM
degradation and mineralization due to temporal changes in redox conditions. The degradation of large NOM to more
bioavailable low molecular weight (LMW) compounds promoted by microbial Mn oxidation, especially in systems
with neutral pH (Jones et al., 2018; Sunda and Kieber, 1994; Ma et al., 2020), is also hypothesized to increase bioa-
vailability of Hg–NOM. However, amendments of Mn oxides were also shown to inhibit Fe, $SO_4^{2-}$ reducing conditions
and thus MeHg formation in sediments (Vlassopoulos et al., 2018).
Hg methylation and mobilization is intensively studied in paddy field soils and peat soils due to their relevance in food
production or the Hg global cycle (Wang et al., 2019; Tang et al., 2018; Liu et al., 2016; Hu et al., 2019; Wang et al.,
2016; Zhao et al., 2018; Zhu et al., 2016; Kronberg et al., 2016; Skyllberg, 2008; Skyllberg et al., 2006). However,
only few studies focused on Hg methylation and mobility in temperate floodplain soils (Frohne et al., 2012; Hofacker
et al., 2013; Gilli et al., 2018; Poulin et al., 2016; Lazareva et al., 2019; Wang et al., 2020; Beckers et al., 2019). As
well, few studies have examined the effect of flooding and/or land use (NOM addition in the form of animal manure)
in polluted soils with respect to Hg release and methylation potential (Tang et al., 2018; Gygax et al., 2019; Zhang et
al., 2018; Hofacker et al., 2013; Frohne et al., 2012). Furthermore, most of these studies were focusing on soils with
rather high OC levels (5 – 10 wt. %) and only few researchers have addressed the decrease of Hg in soil solution of
flooded soils over time, including the fate of colloidal Hg.
This work focused on the effect of the agricultural practices on the Hg mobility and methylation in a real-world con-
taminated fluvisol with specific emphasis on the flooding-draining cycle and manure addition. By conducting micro-
cosm experiments, we studied the effect of these cycles and manure addition on 1.) the release and sequestration of
Hg, 2.) the methylation of Hg and 3.) the evolution of colloidal and particulate Hg in soil solution. The latter was
studied by analyzing different soil solution filter fractions (0.02 and 10 µm) as well as analyzing selected samples by
asymmetric flow field flow fractionation coupled to a $UV_{vis}$ detector, a fluorescence detector and an ICP–MS (AF4–
ICP–MS). Based on the presented state of knowledge, we hypothesise that the manure addition would accelerate the
release of Hg by accelerated reductive dissolution of Mn-oxyhydroxides in these soils and eventually change Hg
speciation in the system towards Hg-NOM complexes and ß-$HgS_{(s)}$ colloids.
**2. Methods and Materials**
**2.1. Sample collection**
We sampled soil from agriculturally used fields in the alpine Rhone Valley in Wallis, Switzerland on September 30[th],
2019. The fields are situated in a former floodplain next to the artificial "Grossgrundkanal" canal. This canal was built
in the 1900s to drain the floodplain and as a buffer for the waste water releases of an chemical plant upstream histor-
ically using Hg in different processes (chlor-alkali electrolysis, acetaldehyde- and vinyl chloride production). The
soils on the floodplain were subjected to Hg pollution from this plant between the 1930s and the 1970s, mostly through
the removal and dispersion of the canal sediments onto the agricultural fields (Glenz and Escher, 2011). After heavy
rain events, the fields are subjected to draining-flooding cycles (Fig. S1) and have been identified as potential hotspots
for Hg methylation and release (Gygax et al., 2019). For this study, soil was sampled from a cornfield and a pasture
field next to the canal. A map and the coordinates of the sampling locations is provided in the supplement (Fig. S1,
Table S1). At each site, a composite sample of approx. 10 kg of soil was sampled between 0 – 20 cm depth from ten
points on the fields. The soil samples were named after their relative pollution and organic carbon levels (High Mer-
cury, Low Carbon (HMLC) and Low Mercury, High Carbon (LMHC), see Part 4.3 below for details on the soils. After
sampling, roots were removed, and the fresh soil was sieved to < 2 mm grain size, further homogenized, split in two
parts and stored on ice in airtight PE Bags for transport to the laboratory. Additionally, approx. 2 L of liquid cow
manure was sampled from a close-by cattle farm. One aliquot of the samples was stored at - 20° C until further pro-
cessing. The remaining part was used for the incubation experiment within 12 h after sampling. A detailed description
of the site and sampling procedures is given in the supplement (Sect. S1).

## 2.2 Microcosm Experiments

An initial incubation was conducted in 10 L HDPE containers in the dark for seven days in an atmosphere of 22 °C and 60 % relative humidity (RH) in order to equilibrate the soils and to prevent a peak of microbial respiration induced by the soil sieving before the onset of the experiment (Fig. 1). After the initial incubation period soils were used in the flooding and draining experiments, which were conducted in 1 L borosilicate glass aspirator bottles (Fig. S2). The environment created through soil flooding in these bottles will be called microcosm (MC) in the following text. Microcosm experiments were performed in experimental triplicate and named after the relative Hg- and organic carbon levels of the used soil (HMLC and LMHC) and the treatment with or without manure addition (added +MNR). The microcosms were equipped with an acid washed suction cup with a pore size of < 10 µm (model: 4313.7/ETH, ecoTech Umwelt-Meßsysteme GmbH, Bonn, Germany). In the following sequence, 700 g of artificial rainwater ($NH_4NO_3$ 11.6 mg $L^{-1}$/ $K_2SO_4$ 7.85 mg $L^{-1}$/ $Na_2SO_4$ 1.11 mg $L^{-1}$/ $MgSO_4 \cdot 7H_2O$ 1.31 mg $L^{-1}$/ $CaCl_2$ 4.32 mg $L^{-1}$) was added to the microcosms. For the manure treatment, 0.6 % (w/w) (3 g) of liquid cow manure was added to the microcosms corresponding to one application of liquid manure on a cornfield following the principles of fertilization of agricultural crops in Switzerland (Richner and Sinaj, 2017) and finally fresh soil was added with a $soil_{dry}$:water ratio of 1:1.4 (w/w) (Fig. S3). Then, the microcosms were gently shaken for at least one minute to remove any remaining air bubbles in the soil and pore space. An additional mixture of fresh soil artificial rainwater (1:1.4 (w/w)) was shaken for 6 h to assess the equilibration of the solid and liquid phase during the experiment. The microcosms were covered with Parafilm®, transferred to the incubation chamber (APT.line™ KBWF, Binder, Tuttlingen, Germany) and incubated in the dark for 14 days in atmosphere of 22 °C and 60 % RH. The incubation temperature was chosen to be close to the daily mean soil temperature in 10 cm depth during summer months between 2015–2019 (21.4 °C) at the closest soil temperature monitoring station (Sion, VS, provided by MeteoSwiss) situated downstream. After the first flooding period, the supernatant water was pipetted off, and remaining water was sampled through the suction cups to drain the microcosms. They were weighted before and after water removal. Then, approximately 25 g of moist soil was sampled by two to three scoops though the whole soils column using a disposable lab spoon. The microcosms were kept drained in an atmosphere of 22 °C and 10 % RH for 14 days. For the second flooding period, the microcosms were again flooded with 500 g of artificial rainwater and incubated for another 14 days in an atmosphere of 22 °C and 60 % RH (Fig. 1). After the incubation, the suction cups were removed, the soils were homogenized and then transferred from the MC to a PE bag and stored at -20 °C until further processing.

## 2.3 Soil and manure characterization

Frozen soil and manure samples were freeze dried to avoid a loss of Hg prior to analyses (Hojdová et al., 2015), ground using an automatic ball mill (MM400, Retsch, Haan, Germany) and analyzed for the following chemical parameters. Carbon (C), nitrogen (N) and sulfur (S) were measured with an elemental analyzer (vario EL cube, Elementar Analysensysteme, Langenselbold, Germany). Organic Carbon (OC) was calculated by subtracting the C concentration of a loss on ignition sample (550 °C for 2 h) from the original C concentration. pH was measured in an equilibrated 0.01 M $CaCl_2$ solution (1:5 soil:liquid ratio). Mineral composition was measured by X-ray diffraction (XRD, CubiX[3], Malvern Panalytical, Malvern, United Kingdom). Trace and major metals (e.g. Fe, Mn, Cu) and Hg were extracted

from soils using a 15.8 M nitric acid microwave digestion and measured using an Inductively Coupled Plasma - Mass
Spectrometer (ICP–MS, 7700x, Agilent Technologies, Santa Clara, United States of America). Methylmercury was
selectively extracted with HCl and dichloromethane (DCM) using an adapted method described elsewhere (Gygax et
al., 2019). We modified this method to achieve high throughput (64 Samples per run) and measurements by High
Pressure Liquid Chromatography (HPLC, 1200 Series, Agilent Technologies, Santa Clara, United States of America)
coupled to the ICP–MS. Details on laboratory materials, extractions, analytical methods and instrumentation are pro-
vided in the supplement (Sects. S2, S3). The change in MeHg concentration in the microcosms were likely a result of
the simultaneous production and degradation of MeHg. Thus, the term "net MeHg production" was used to represent
these processes. We calculated the relative net MeHg production during the incubation as the relative difference of
MeHg/Hg ratios between two time points (t) using Eq. (1).
$net\ MeHg\ production\ (\%) = \dfrac{\left(\frac{MeHg}{Hg}_{t_{i-1}} - \frac{MeHg}{Hg}_{t_i}\right)}{\frac{MeHg}{Hg}_{t_{i-1}}} \times 100$        (1)

### 185  2.4 Soil description

Both soils were identified as *Fluvisols gleyic.* They have a silt loam texture, the same mineral composition but differing
Hg and organic carbon ($C_{org}$) concentrations (Table 1). For elements relevant for Hg cycling, Hg molar ratios (Hg:Cu,
Hg:$C_{org}$, Hg:Mn) differ between samples and soils used in similar incubation experiments (Hofacker et al., 2013;
Poulin et al., 2016). We note that the $[C_{org}/Mn]_{molar}$ was 30 % higher in the LMHC soil compared to HMLC. X-Ray
diffractograms of both soils are shown in Fig. S4. The soils diffractograms are overlapping each other and the quali-
tative analyses of the diffractograms show that the soils parental material is composed of the same five main mineral
phases, quartz, albite, orthoclase, illite/muskovite, calcite.

### 193  2.5 Soil solution sampling and analyses

Soil solution was sampled 0.25, 1, 2, 3, 4, 5, 7, 9, 11, 14 days after the onset of each flooding period respectively (Fig.
1, Fig. S5). It was sampled though the tubing connected to the suction cup (< 10 µm pre size). The first 2 ml were
sampled with a syringe and discarded to prime the system and condition the tubing. After, 4 ml were drawn through
an airtight flow-through system to measure the redox potential (Hg/HgCl ORP electrode) and pH. Then, approximately
35 ml of soil solution were sampled using a self-made syringe pump system allowing for a regular flow and minimal
remobilization of fine particles. Like this, 4-6 % of the added artificial rainwater volume was sampled at each sampling
point (Fig. S3). Throughout the experiment the soils remained entirely submerged. At each sampling time, sample
splits were preserved without further filtration (<10 µm) and filtered at 0.02 µm (Whatman® Anodisc 0.02 µm, Sigma-
Aldrich, St. Louis, United States of America). Additionally, at 2,5 and 9 days an additional sample split was filtered
at 0.45 µm (Polytetrafluoroethylene Hydrophilic, BGB, Boeckten, Switzerland) for colloid characterization. Incuba-
tion experiment blanks were taken by sampling MilliQ water through from an empty 1 L borosilicate aspirator bottle
3 times throughout the experiment. Subsequently, the samples were subdivided and treated for different analyses.
They were preserved in 1 % $HNO_3$ for multi elemental analysis (Mn, Fe, Cu, As) and in 1 % $HNO_3$ and 0.5 % HCl
for Hg analysis and analyzed by ICP–MS. For major anion ($Cl^-$, $NO_3^-$, $SO_4^{2-}$) and cation ($K^+$, $Na^+$, $Mg^{2+}$, $Ca^{2+}$) meas-
urements, samples were diluted 1:4 in ultra-pure water and analyzed by Ion Chromatography (Dionex Aquion™,
Thermo Fisher Scientific Inc., Waltham, United States of America). Samples for Dissolved Organic Carbon (DOC),
Particulate Organic Carbon POC and Total Nitrogen Bound ($TN_b$) were diluted 1:5 and stabilized using 10 µl of 10
% HCl and measured using an Elemental Analyzer (vario TOC cube, Elementar Analysensysteme, Langenselbold,
Germany). Incubation experiment blanks were below 4.75 mg $L^{-1}$ and 22.4 µg $L^{-1}$ for DOC and $TN_b$, respectively.
These relatively high blank values might originate from either the syringes or the suction cups (Siemens and
Kaupenjohann, 2003). Uncertainties of soil solution parameters are displayed as 1SD of the triplicate incubation ex-
periments throughout the manuscript. $HCO_3^-$ concentrations were estimated based on the ionic charge balance of the
soil solution using VisMinteq (https://vminteq.lwr.kth.se/). A detailed schedule and list of analyses is provided in
Figure 1. Concentrations of specific filtered fractions are labelled with subscripts (e.g. $HgT_{<0.02µm}$) for all measured
metals. Particulate concentrations (0.02 µm < X < 10 µm) (e.g. P-Fe) and its proportion to the total (e.g. $P-Mn_{rel}$) were
determined as the difference between unfiltered and filtered concentration (Table 2).

## 2.6 Characterization of Colloids (AF4)

An aliquot of the soil solution was used for characterization of colloids in one out of three replicate microcosms (Rep1)
of each treatment on days 2, 5, 9 days after the onset of each flooding period respectively. Right after sampling, the
aliquots were transferred to a $N_2$ atmosphere in a glove box. There, the samples were filtered to < 0.45 µm and pre-
served in airtight borosilicate headspace vials at 4 °C. Colloidal size fractions and elemental concentrations of the
filtrates were analyzed by Asymmetrical Flow Field-Flow Fractionation (AF4, AF2000, Postnova analytic, Landsberg
am Lech, Germany) coupled to a $UV_{254nm}$ absorbance detector (UV, SPD-M20A, Shimazu, Reinach, Switzerland), a
Fluorescence detector (FLD, RF-20A, Shimazu, Reinach, Switzerland) and an ICP–MS (7700x, Agilent Technologies,
Santa Clara, United States of America) within 14 days after sampling. Colloids contained in 1 mL of samples were
separated in a channel made of a trapezoidal spacer of 350 µm thickness and a regenerated cellulose membrane with
a nominal cut-off of 1 kDa used as accumulation wall. The mobile phase used for AF4 elution was 10 mM $NH_4NO_3$
at pH 7 and was degassed prior entering the channel by argon flowing. A linear decrease of crossflow from 2 to 0 mL
$min^{-1}$ over 20 min was used after injecting the samples at an initial crossflow of 2.7 mL $min^{-1}$. At the end of a run, the
crossflow was kept at 0 mL $min^{-1}$ for 5 min in order to elute non-fractionated particles. Retention times were trans-
formed into hydrodynamic diameters ($d_h$) by an external calibration using Hemocyanin Type VIII from Limulus pol-
yphemus hemolymph (monomer $d_h = 7$ nm, Sigma-Aldrich) and ultra-uniform gold nanoparticles (Nanocomposix) of
known $d_h$ (19 nm and 39 nm). Additionally, the elution of the smallest retention times ($d_h < 10$ nm) were converted
into molecular masses (Mw) using PSS standards (Postnova analytic, Landsberg am Lech, Germany) with Mw ranging
from 1.1 to 64 kDa (Fig. S6), using $AF4-UVD_{254nm}$.
Fractograms obtained in Counts Per Seconds (CPS) from Time Resolved Analysis (TRA) acquisition were converted
to µg $L^{-1}$ using external calibrations made from a multi-element standard solution (ICP multi-element standard solution
VI, Merk, Darmstadt, Germany) diluted in 1 % $HNO_3$ or a Hg standard (ICP inorganic Hg standard solution, Trace-
CERT®, Sigma-Aldrich, St. Louis, United States of America) diluted in 1.0 % $HNO_3$ and 0.5 % HCl. The different

size fractions were obtained by multiple extreme-shaped peak fitting, using OriginPro 2018 software (OriginLab Corporation). The peaks obtained were then integrated individually, after conversion of elution time to elution volume, to provide the quantity of Hg in each size fractions (Dublet et al., 2019). The analytes passing the 1 kDa membrane are considered as the (< 1 kDa) truly dissolved fraction. It was calculated by subtracting the concentrations of colloidal HgT recovered by AF4–ICP–MS (total integration of the Hg signals) to the total dissolved HgT concentrations measured separately by ICP–MS in corresponding acidified samples. The concentration of truly dissolved Hg is displayed as $HgT_{<1kDa}$ for the rest of the article (Table 1). AF4–ICP–MS, $UV_{254nm}$ and fluorescence signals were used to further characterize Hg bearing colloids, after hydrodynamic size separation by AF4. The $UV_{254nm}$ light absorption is widely used to detect organic compounds but it should be noted that part of the $UV_{254nm}$ light signal can as well originate from Fe(II) or Fe hydroxides (Dublet et al., 2019). This was not the case in this study since $UV_{254nm}$ signals co-eluted with C signals recorded by ICP–MS and matched the fractograms obtained by the FLD detector tuned at the wavelengths specific for humic-like fluorophores. It is therefore assumed that $UV_{254nm}$ signal represents organic compounds throughout the manuscript.

## 3. Results

### 3.1 Soil solution chemistry and Hg dynamics

In the HMLC microcosms, the pH of the soil solutions remained in a neutral to alkaline range of 8.0 to 8.4 during the incubation experiment (Fig. S7). Soil solution conditions and concentrations of constituents support a continuous reduction of soils with increased flooding time (Fig. 2a). Soil solution $NO_3^-$ depletion was observed during the first 7 days of incubation (Fig. 2b). Nitrate was under detection limit for the second flooding phase. At day 7, Mn concentrations increased together with a marginal increase of Fe (Fig. 2c-f). This was coincided with a decrease of the relative particulate fraction (P-$Mn_{rel.}$ and P-$Fe_{rel.}$) of these metals. Release of Mn and Fe were assumed to mark the onset of reductive dissolution of Mn- and Fe-oxyhydroxides. The decrease in sulphate ($SO_4^{2-}$) concentration could not be used to assess the onset of of sulphate reduction. This is due to a chemical gradient between supernatant water and soils solution demonstrated by the continuous decrease in concentration of conservative ions ($Cl^-$, $Na^+$, $K^+$) (Sect. 4.4). To monitor sulphate reduction, we use the molar ratios of $SO_4^{2-}$ to $Cl^-$ (Fig. 2g). Sulphate to chloride ratios stood constant during the first flooding and slightly increased at the onset of second flooding phase. This suggests that no sulphate reduction took place in the HMLC microcosms during the whole experiment. The DOC concentration ranged between 37.5 and 106 mg $L^{-1}$ (Fig. 2h). Both $HgT_{<0.02\mu m}$ and $HgT_{<10\mu m}$ concentrations remained low between day 0-5 (Phase 0), then increased together with the Mn release between days 5-11 (Phase 1) and decreased between 14-29 (Phase 2) during the draining period (Fig. 3a). The relative fraction of particulate HgT (P-$HgT_{rel.}$), gradually decreased from a maximum of 88 % to a minimum of 25 % during phase 0 and phase 1, but increased again to 60-77 % during phase 2 (Fig. 3b-c). $Cu_{<0.02\mu m}$ concentrations increased up to $88.2 \pm 17.5 \mu g\ L^{-1}$ within the first 4 days and then gradually decreased to $30.6 \pm 3.54\ \mu g\ L^{-1}$ at day 14 (Fig. 4a). Arsenic concentrations simultaneously increased with the release of Fe during the whole incubation (Fig. 4b).

During the second flooding period, individual microcosms behaved differently in the HMLC run. The differences of soil solution $E_h$ and redox sensitive metals (e.g. Mn, Fe, Hg, Cu) were apparent from the start of the second flooding (Figs. 2c-f, 3a-c, 4a). Contrastingly, DOC concentrations and pH remained similar between incubators (Figs. 2h, S7). One replicate (Rep1) showed a pronounced increase of redox potential after the draining period (Fig. 2a). The $E_h$ remained high (150 to 300 mV) for the whole second flooding period. A depletion and subsequent release of Mn in soil solution was observed, indicating the formation and redissolution of Mn oxyhydroxide minerals (Fig. 2c-d). Subsequently, $Mn_{<0.02\mu m}$ increased and peaked at 448 µg $L^{-1}$ by the end of the experiment in Rep1. The $E_h$ of Rep2 was lower (between 28 and 120 mV), Mn concentrations did not decrease during the draining phase, and a release of Fe was observed during the second flooding phase indicating the reduction of Fe oxyhydroxides. Rep3 had a $E_h$ in the range of Rep2 but neither a rerelease of Mn nor a release of Fe was observed during the second flooding phase. Also, HgT behaved differently within incubators during the second flooding period. Between days 29-42 (Phase 3), $HgT_{<0.02\mu m}$ and $HgT_{<10\mu m}$ concentrations increased or remained at higher levels for Rep1 and Rep3. During this phase P-$HgT_{rel}$ vastly decreased and was at a minimum of 1-7 % by the end of the incubation. Contrastingly, $HgT_{<0.02\mu m}$ and $HgT_{<10\mu m}$ stayed constantly low for Rep2 during phase 3 and P-$HgT_{rel}$ remained overall above 50%. The Rep1 was the only MC that showed an increase in Cu concentrations during the draining phase (Fig. 4a).

In the HMLC +MNR microcosms, pH remained in the range of 8 to 8.35 with minor fluctuations over both flooding periods (Fig. S7). The redox potential decreased rapidly from approx. $E_h$ 300 mV to $5.27 \pm 14.4$ mV within the first 14 days and remained constant at $14.3 \pm 8.12$ mV during the second flooding period. Depletion of $NO_3^-$ was observed within the first day of incubation and was under detection limit during the second flooding period (Fig. 2b). A rapid release of Mn started at day 2 and a slow release of Fe started at day 3 of first flooding period (Fig. 2c-f). The $[SO_4^{2-}]:[Cl^-]$ ratios decreased from $0.57 \pm 0.01$ to $0.37 \pm 0.02$ between day 4-29. During the second flooding period $[SO_4^{2-}]:[Cl^-]$ ratios initially increased slightly between day 29-31 and then decreased to a minimum ($0.12 \pm 0.05$) by the end of the incubation (Fig. 2g). DOC concentrations were between 72.2 and 134 mg $L^{-1}$ (Fig. 2h). This was significantly higher (3 to 43 mg $L^{-1}$) than in HMLC without manure. In these microcosms $HgT_{<0.02\mu m}$ and $HgT_{<10\mu m}$ concentrations instantly increased together with the Mn release between days 0-4 (Phase 1) decreased during the days 5-14 (Phase 2) and remained low between day 14-42 (Phase 3) (Fig. 3 a-c). The particulate HgT (P-$HgT_{rel.}$) decreased to 30-52.5 % in phase 1 and remained overall above 50 % for the rest of the incubation. At the onset of phase 2 black precipitates were visually observed in the HMLC +MNR microcosms (Fig.S13). Cu concentrations decreased gradually during the course of the incubation experiment (Fig. 4a). Arsenic concentrations simultaneously increased with the release of Fe during the whole incubation (Fig. b).

LMHC differed from HMLC in soil solution chemistry. In both treatments (LMHC and LMHC +MNR), pH remained neutral but gradually decreased from 8.2 to 7.5 during the incubation (Fig. S7). Soil reduction progressed rapidly from a max of 332 mV at day 3 to -14.3 mV at day 14 (Fig. 5a). During the second flooding $E_h$ stayed in the range of - 2.3 to 34.5 mV. Nitrate was exhausted within the first day of incubation and marked the onset of Mn release. Mn as well as DOC concentrations gradually increased during the first flooding period (Fig. 5b-c). Fe release started on day 4 and day 6 in LMHC and LMHC +MNR respectively (Fig. 5d). A decrease in $[SO_4^{2-}]:[Cl^-]$ ratio was observed after day 5

and remained stable at $0.03 \pm 0.04$ during the second flooding period. This is indicative for sulphate reduction during the draining phase and the second flooding phase (Fig. 5e). Soil solution $HgT_{<0.02\mu m}$ concentration ($25 - 160$ ng $L^{-1}$) were two orders of magnitude lower than in the HMLC runs (Fig2. 3a,6a). Dissolved $HgT_{<0.02\mu m}$ degreased during the first flooding period (phase 1), increased during the draining period (phase 2) and gradually decreased again during the second flooding period (phase 3) (Fig. 6a-c). No other soil solution parameter followed the trend of $HgT_{<0.02\mu m}$. Particulate $HgT_{<10\mu m}$ decreased during phase 1 and remained low during phase 2 and 3. In the LMHC microcosms P-$HgT_{rel.}$ changed drastically between phase 1 ($> 65$ %) and phase 3 ($<< 50$ %) (Fig. 3c). In the LMHC +MNR micro-cosms the P-$HgT_{rel.}$ was high during the phase 1 ($> 65$ %) and fluctuated between phase 3 ($<< 50$ %) (Fig. 3c). Cu concentrations gradually decreased during the course of the experiment (Fig 7a). Arsenic concentrations simultane-ously increased with the release of Fe during the whole incubation (Fig 7b).

## 3.2 Colloidal Hg (AF4)

Hg bearing colloids were detected in all soil solution samples of HMLC incubations. Due to low signal to noise ratios ($< 3$) we did not detect colloidal Hg in samples of the LMHC incubations. Figure 8 shows the evolution of concentra-tions and relative proportions of HgT size fractions. Generally, changes in proportions were apparent during phases of Hg release and decrease in soil solution, but little change was observed during when Hg concentrations were stag-nant (HMLC +MNR, Phase 3). The proportion of truly dissolved $HgT_{<1kDa}$ varied between 0 % and 67 % in the HMLC experiment and was high during Hg release to soil solution (phases 1 and 3) (Fig. 8). In the HMLC +MNR treatment, $HgT_{<1kDa}$ were lower and ranged between 0 % and 29 %. The colloidal Hg can be divided into 3 main fractions (Fig. 9). The first Hg colloidal fraction showed a main peak ranging between $1 - 40$ kDa ($d_h < 6$ nm) and was associated with $UV_{254nm}$-absorbing compounds and various metals (Mn, Fe, Cu, Ni, Zn). This fraction was interpreted as humic substance type Hg–NOM. The proportion of this colloidal Hg fraction varied with no specific trends from 11.5 to 23.3 % in HMLC and 13.6 to 38.6 % in HMLC +MNR throughout the course of the experiment. A second fraction of Hg colloids ranged between 6 nm and 20 nm. This well-defined size fraction was eluting in the tail of the first fraction for other metals (e.g. Fe, Mn, Cu) but did not overlap with $UV_{254nm}$ and fluorescence signals (Fig. 9). This fraction could not be chemically defined but is hypothesized to consist of ß-$HgS_{(s)}$ colloids. In the HMLC run, we observed a decrease in the proportion of these inorganic colloids from 28 % in phase 0 to 15.3 % at the end phase 3 (Fig. 9). In the HMLC +MNR treatment, the proportion of this fraction ranged between 29.5 % and 41.9 % during the phases 1 and 2 and could not be detected during the phase 3. Further, we observed a third colloidal fraction that continued to elute after the stop of the AF4 crossflow and it included colloids in the range of $30 - 450$ nm (effective cut-off of the filter used for the sample preparation). In some cases, this fraction was better fitted using two overlapping populations (Fig. 9, Figs. S9-S12). In all the cases, HgT signal was associated with those of other metals and a slight bump of the $UV_{254nm}$ signal but more specifically an increase of fluorescence signal associated to protein-like fluorophores. This fraction decreased continuously in the HMLC runs during the incubation from 32.4 %  in phase 2 , to 5.6 % in phase 2 and stood under 9.1 % during phase 3. By contrast, the HMLC +MNR showed an increase in the proportion of this fraction from 7.3 % in phase 1 to 25.3 % by the end of phase 3 (Fig. 8). The deconvolution of the fractograms included an intermediate fraction of Hg bearing colloids ranging between $d_h = 6$ nm and $d_h = 450$ nm depending on the sample.

This fraction was added to refine the fractogram fittings but could not directly be associated to another measured
metal. This indicates that this population represents a polydispersed Hg particle population although in some cases the
presence of small Hg particles dominates. This broad fraction was not detected in HMLC +MRN treatments during
phases 1 and 2 but made up > 30 % during phase 3.

### 3.3 Net MeHg production in soil.

Soil MeHg levels fluctuated over the course of the incubation experiment (Fig.10 and Table 2). Highest net MeHg
production was observed during the first flooding period for the treatments with manure (up to + 81 %) and during
the draining phase for the treatments without manure (up to + 73.1 %). We observed a significant decrease of
MeHg/HgT and absolute MeHg concentrations in all incubators during the second flooding period (Fig. 10). In all
microcosms, MeHg/HgT increased by a factor of 1.18 to 1.36 throughout the incubation (Table 2).

## 4. Discussion

### 4.1 Mercury release and sequestration.

Cornfield soil (HMLC) and pasture field soil (LMHC) behaved differently in this incubation experiment and will be
discussed separately. In the cornfield soil (HMLC) Hg and Mn releases were simultaneous and started when soil
solution $E_h$ entered the field of Mn reduction below approx. 300mV (Figs. 2c,3a), strongly suggesting that this Hg
pool was released by reductive dissolution of Mn-oxyhydroxides. During all experiments, low Hg:DOM ratios (<<1
nmol Hg (mg DOM)$^{-1}$) suggest that strong binding sites of DOM were never saturated with respect to mercury, as-
suming a binding site $[RS_2^{2-}]$ density of 5 nmol Hg (mg DOM)$^{-1}$ and that DOC is 50 % the DOM (Haitzer et al., 2002).
The low Hg:DOM ratio suggests that Hg is mainly present as complexed with DOM given reported strong interaction
with thiol sites of DOM. However, these assumptions might not reflect the actual composition of DOM which might
drastically differ in amended soils (Li et al., 2019). Reductive dissolution of Mn-oxyhydroxides drives both 1.) the
release of labile Hg-NOM complexes and $Hg^{2+}$ sorbed on the oxide's surfaces and/or 2.) enhanced the degradation
and mineralisation of unsubtle NOM binding Hg in soils (Jones et al., 2018). After Hg release (phase 1), Hg concen-
trations remained high and the relative particulate Hg fraction was low throughout the experiment. This illustrates that
the released Hg-pool mainly originated from Mn-oxyhydroxides or degradation of suspended POM during Mn reduc-
tion. However, the released Hg-pool is relatively small compared the HgT levels of the soil. We estimate that about
12.8 ± 4.2 µg kg$^{-1}$ Hg (0.02 % of HgT$_{soil}$) was evacuated by sampling during the experiment. In this fluvisol, Hg
mobilization is thus mainly driven by reductive dissolution of Mn oxyhydroxides. Direct mobilization of DOM was
reportet to govern Hg levels in peat soils, Histosols or Podsols in boreal environments (Åkerblom et al., 2008;
Kronberg et al., 2016; Jiskra et al., 2017) or floodplain soils with higher OC levels (Beckers et al., 2019; Wang et al.,
2021) in temperate soils.
Further, Hg mobilisation was not simultaneous to Cu release. This was reported for polluted soils with high Cu levels
(Hofacker et al., 2013) and comparably low Hg/Cu$_{molar}$ ratio in the soil matrix. In neighbouring soils, the main Hg
pool was previously reported as HgS$_{(s)}$ and Hg complexed by recalcitrant NOM (Grigg et al., 2018). Earlier studies

assumed that 0.1 to 0.6 % (w/w) of NOM was reduced sulphur with high affinity to Hg (Grigg et al., 2018; Ravichandran, 2004). Following this assumption, reduced sulphur groups of the cornfield soils NOM could sorb between 11.9 to 71.9 mg kg$^{-1}$ of Hg. The soils high Hg concentration ($47.3 \pm 0.5$ mg kg$^{-1}$) suggests that soil NOM thiol sites are likely saturated in terms of Hg. Therefore, saturated NOM sorption sites are not competing with Mn-oxyhydroxide sorption sites, resulting in a substantial Mn-oxyhydroxide bound Hg-pool. This leads to a higher mobility of Hg upon reductive dissolution of Mn-oxyhydroxide compared to fluvisols used in other incubation studies (Hofacker et al., 2013; Poulin et al., 2016; Beckers et al., 2019).

During the second flooding phase, the cornfield soil (HMLC) runs showed a higher variability in redox sensitive soil solution parameters (Fig. 2). This might be explained as 1.) a shift in microbial communities, 2.) disturbance of the soil column by invasive soil sampling in between the flooding periods or 3.) uneven draining of the pore space after the first flooding. It can also reflect how redox cycle can be easily affected *in situ*. We suggest that the second release of Mn and Hg in Rep1 is due to Mn re-oxidation during the draining period and a second reductive dissolution of Mn oxyhydroxides upon reflooding. This is supported by the elevated $E_h$ at the onset of the second flooding. Further, Mn reduction oxidation and reduction cycles were shown to enhance the degradation of NOM to more labile forms (Jones et al., 2018) which might contribute to the degradation/mineralization of recalcitrant Hg-NOM. The HMLC Rep3 showed a second release of Hg without a remobilization of Mn. Changing redox conditions have been shown to enhance microbial respiration and therefore NOM degradation (Sunda and Kieber, 1994). Thus, we interpret the second Hg release in Rep 3 as a degradation/mineralization of NOM that bound Hg.

The carbon amendments were reported to decrease total Hg release in polluted floodplain soils (Beckers et al., 2019) but may have a mobilizing effect in NOM depleted environments (Eckley et al., 2021). The addition of manure accelerated the release of Hg through reductive dissolution of Mn oxyhydroxides in the cornfield soil (HMLC). Mercury was released 4 day earlier, as result of additional labile carbon of the liquid manure 1.) acting as electron donor enhancing microbial soil reduction (Liu et al., 2020), 2.) act directly as reductant of the Mn oxyhydroxides (Remucal and Ginder-Vogel, 2014). In the manure treatment, we observed a fast decrease of Hg concentration and a constantly high proportion of particulate P-HgT$_{rel.}$ even after the plateau of Mn concentration in soil solution and the relative decrease of particulate Mn. The addition of manure a source of POM (manure was sieved to < 500 µm) and increased DOC approximately by 20 mg L$^{-1}$. Sorption of Hg is directed towards thiol rich high molecular weight NOM (Liang et al., 2019) following different ligand exchange reactions (e.g. carboxyl-groups to thiol groups) which happen within days (Miller et al., 2009; Chiasson-Gould et al., 2014). The constant of P-Hg$_{rel}$ proportion is suggested to be partly caused by the complexation of dissolved Hg with the added POM of the manure.

In addition, we visually observed black precipitates (Fig. S13) and the decrease of [SO$_4^{2-}$]:[Cl$^-$] ratios (Fig. 2g) at the onset of Hg decrease (phase 2) in the microcosms with manure addition. This indicates the precipitation of sulphide mineral particles. Although, redox potential measurements did not indicate sulphate reduction, the monitoring of $E_h$ in soil solution provides only a qualitative measure in a complex soil systems. We suggest that, formation and aggregation of ß-HgS$_{(s)}$ explains the faster decrease in the manure amended experiment. Furthermore, formation of meta-cinnabar ß-HgS$_{(s)}$ was observed under oxic conditions by conversion of thiol bound Hg(SR)$_2$ (Manceau et al., 2015). The formation and aggregation of ß-HgS$_{(s)}$ is further supported by AF4 results (Sect. 4.2).

Click or tap here to enter text.Hofacker et al. (2013) reported a quantitatively relevant incorporation of Hg into metallic
$Cu^0$ particles. However, we do not consider this a relevant pathway, due to the relatively high Hg/Cu$_{molar}$ ratio in our
soil compared to Hofacker et al. (2013). Although the simultaneous decrease of Hg and Cu may be interpreted as the
immobilization of Hg though incorporation into metallic Cu particles, i) we did not observe the formation of colloidal
Cu associated with Hg (Sect. 6.2) and ii) relatively high Hg/Cu molar ratios indicate that the decrease of Hg in the soil
solution cannot be solely explained by this mechanism as Hg would be marginally incorporated metallic $Cu^0$ particles.
As well, Hg in soil solutions is volatilized by reduction of $Hg^{2+}$ to $Hg^0$ (Hindersmann et al., 2014; Poulin et al., 2016;
Li et al., 2021). Our experimental design did not allow for quantification of gaseous $Hg^0$ and it may have exited the
microcosms since they were only sealed with parafilm. Reduction of $Hg^{2+}$ may happen both biotically (Grégoire and
Poulain, 2018) and abiotically under UV-light and in the dark (Allard and Arsenie, 1991). Biotic reduction is a detox-
ication mechanism of bacteria carrying *merA* genes in Hg polluted environments. Biotic volatilization has been ob-
served in neighboring soils of our sampling site (Frossard et al., 2018). Organic amendments and high Hg levels have
been shown to increase the abundance of Hg reducing bacteria (Hu et al., 2019). Further, dark abiotic reduction of
$Hg^{2+}$ complexed to functional groups of DOM in soils has been demonstrated (Jiang et al., 2015). However, it is
unlikely that Hg reduction can solely explain the decrease of Hg in the soil solution in our microcosms. We therefore
interpret the decrease in Hg concentration to be due to a combination of manure NOM complexation and sequestration
together with the formation of HgS$_{(s)}$ during flooding. Our data shows that manure addition may have an immobilizing
effect on Hg in flooded soils. By contrast, carbon amendments may increase Hg mobility and methylation in NOM
depleted and cinnabar rich mountain soils (Eckley et al., 2021).

In the pasture field soil (LMHC), soil solution Hg concentrations remained at low levels ($< 0.16$ µg L$^{-1}$ Hg$_{<0.02\mu m}$)
during the whole experiment in both treatments (Fig. 6a). Unlike in the cornfield soil (HMLC), we did not observe a
simultaneous release of Hg upon Mn reduction (Fig. 5c). We explain this with the not completely Hg saturated NOM
in this soil, if we assume that $0.1 - 0.6$ % (w/w) of NOM was reduced S with high affinity to Hg (Grigg et al., 2018;
Ravichandran, 2004; Skyllberg, 2008). Thus, the pasture field soil has a rather limited pool of labile Hg compared to
the cornfield soil. Both Hg$_{<0.02\mu m}$ and Hg$_{<10\mu m}$ negatively correlate with the sum of sampled soil solution ($R^2 = -0.841$,
p= <0.001) during both flooding periods and fastly decreased. This suggests that the concentration gradient between
supernatant artificial rainwater and the soil solution contributed to the fast exhaustion of the small labile Hg pool in
pasture field soil. The presence of this concentration gradient in our incubation setup is confirmed by the continuously
decreasing concentrations of conservative ions (Cl$^-$, Na$^+$, K$^+$) in soil solutions of the HMLC runs (Sect. S5.2, Figs. S7,
S8). The relatively high proportion of particulate Hg vastly decreased during the draining period (Fig. 3b,c) and we
speculate that this change is a result of the mobilization of the POM–Hg pool by mineralization/degradation of NOM
which sorbed Hg during the draining period (Jones et al., 2018). In summary, flooding of the pasture field soils did
mobilize only a small pool of particulate bound Hg which was exhausted within the first flooding period.

## 4.2 Colloidal Hg

For runs without manure, AF4 results show that the Hg released from Mn-oxyhydroxides (Sect. 6.1.2) was dominated by truly dissolved Hg ($Hg^{2+}$ or LMW–NOM–Hg) (Fig. 8). The high $Cl^-$ concentrations (up to 800 mg $L^{-1}$, Fig. S14) likely influenced the Hg speciation in the soil solution, as chloride is a main complexant for $Hg^{2+}$ (Li et al., 2020; Gilli et al., 2018). During Hg release, the proportions of larger Hg colloids (> 25 nm) decreased. The stable proportion of humic substances bound Hg and inorganic Hg colloids between 6 nm and 25 nm indicates that once released no major adsorption or aggregation of truly dissolved Hg and larger colloidal Hg occurs. Additional complexation of Hg by DOM can be excluded if we assume the saturation state of thiol-sites of the NOM pool in the soil (Sect. 6.1.2). These observations illustrates the remarkably high Hg mobility and potentially increased bioavailability (proportion of truly dissolved Hg) to Hg metabolizing microorganisms compared to other studies (Hofacker et al., 2013; Poulin et al., 2016). These authors did either not observe Hg in truly dissolved form or a decrease to low levels within the first days of incubation. Overall, the released Hg from cornfield soil (HMLC) shows a high mobility and might represent a possible threat to downstream ecosystems and a source for Hg methylating bacteria. However, the total Hg released and sampled from soil solution represents a rather small pool (12.8 ± 4.2 µg HgT $kg^{-1}$ soil) of the total Hg (47.3 ± 0.5 mg $kg^{-1}$). Further work would be needed to establish a Hg flux model to better understand *in situ* soil Hg mobility in these soils.

The manure addition had a key effect on the proportions of colloidal fractions in soil solution, and overall led to a low proportion of truly dissolved fraction (Fig. 8). We suggest that the distinct fraction of colloids with $d_h$ = 6 – 25 nm represents metacinnabar like $HgS_{(s)}$ colloids (Gerbig et al., 2011). This is supported by the onset of sulphate reduction in phase 2 (Rivera et al., 2019; Poulin et al., 2016) and reported Hg-NOM interactions that may cause the precipitation of Hg bearing sulphide phases ($FeS_{(s)}$, ß-$HgS_{(s)}$) (Manceau et al., 2015) (Sect. 6.1.1). The size of ß-$HgS_{(s)}$ nano particles formed from free sulphide is dependent in the sulphide concentration as well as on the Hg:DOM ratio (Poulin et al., 2017). The formation of a distinct size fraction of HgS(s) has experimentally observed at comparable Hg:DOM ratios (Gerbig et al., 2011). The Hg colloidal distribution was dominated by the presence of large fractions ($d_h$ = 30 - 450 nm). Larger organic acids with high aromaticity usually contain higher proportions of thiols groups than smaller molecules and selectively complex Hg (Haitzer et al., 2002). This suggests that Hg complexation is kinetically driven and it can shifts from LMW–DOM to larger NOM and larger aggregates of POM as supported by earlier incubation experiments (Poulin et al., 2016). We therefore interpret that the relative increase of Hg colloids with $d_h$ = 30 – 450 nm (Fig. 8) is caused by 1.) complexation of the released dissolved $Hg_{<1kDa}$ by strong binding sites of thiol rich NOM in larger clay-organo-metal complexes and 2.) the aggregation of $HgS_{(s)}$ colloids during the experiment. Although the presence of e.g. humic substances and larger NOM was shown to narrow the size range of $HgS_{(s)}$ nanoparticles precipitating from solution (Aiken et al., 2011) , through time, these colloids may grow, aggregate and form clusters in a wide size distribution (Deonarine and Hsu-Kim, 2009; Poulin et al., 2017). Thus, their aggregation during the draining period may explain the decrease in monodisperse Hg bearing colloids, also leading to sequestration of Hg in the soil matrix, without remobilization during the second flooding. Our data suggests meta cinnabar formation (ß-$HgS_{(s)}$) in a distinct size fraction ($d_h$ = 6 – 25) and their aggregation to large fractions ($d_h$ = 30 – 450 nm) at environmental conditions in real-world samples.

**4.3 Net MeHg production in soil.**

The studied soils show uncommonly high initial MeHg levels (6.4 – 26.9 µg kg$^{-1}$) when compared to other highly polluted mining or industrial legacy sites (Horvat et al., 2003; Neculita et al., 2005; Qiu et al., 2005; Fernández-Martínez et al., 2015), supposedly as a result of a flooding event prior to sampling resulting in a net MeHg production. Still, we observed significant net MeHg production during the first 28 days of the incubation resulting in even higher MeHg concentrations of up to 44.81 µg kg$^{-1}$ (Table 3; Fig. 10). Soils treated with manure showed a faster net MeHg production with highest increase of MeHg during the first flooding period. Controls showed highest net MeHg production during the draining period and reached similar levels of MeHg at the start of the second flooding on day 28 (Fig. 10). For cornfield soil (HMLC), both treatments show a high concentration of bioavailable Hg$^{2+}$ or Hg associated with labile NOM (HgT$_{<0.02\mu m}$ > 15µg L$^{-1}$) in soil solution during the first flooding. Net MeHg production is therefore rather limited by cellular uptake of Hg or the microbial activity of methylating microorganisms than bioavailability. Thus, we interpreted the addition of labile carbon in the form of manure to result in a higher microbial activity and net MeHg production during the first flooding period. However, we did neither assess the activity nor the abundance of Hg methylating bacteria such as sulphate reducers (SRB), Fe reducers (FeRB), archaea or firmicutes (Gilmour et al., 2013). In the runs without manure addition, a substantial part of Hg was methylated during the draining period. This indicates that even if low concentrations of Hg is released (LMHC microcosms day 14: HgT$_{.<0.02\mu m}$ < 50 ng L$^{-1}$) a substantial amount of Hg can be methylated. Micro- and meso pore spaces with steep redox gradients act as ideal environments for microbial methylation even in drained and generally aerobic system (e.g. HMLC without manure during the draining period).

Further, we observed a decrease in absolute MeHg concentrations in all microcosms during the second flooding period. Oscillating net de-/methylation in environments characterized by flood-drought-flood cycles have been reported earlier (Marvin-DiPasquale et al., 2014). Degradation of MeHg was reported to happen either abiotically by photodegradation or biotically by chemotrophic reductive or oxidative demethylation by microorganisms carrying the *mer*-operon (Grégoire and Poulain, 2018). Photodegradation of MeHg can be excluded as the experiment was conducted in the dark. However, demethylation could have happened as biotic reductive demethylation. A possible explanation is a MeHg detoxification reaction by microorganisms carrying the *mer*-operon (*merB*) (Hu et al., 2019; Frossard et al., 2018; Dash and Das, 2012). However, we can only hypothesize about demethylation mechanisms, as neither communities (DNA) nor gene expression (mRNA) dynamics in the soils were analysed during the experiment.

**4.4 Experimental Limitations**

Incubation experiments on a laboratory scale are a common way to study the changes in mobility of trace elements in floodplain soils (Gilli et al., 2018; Frohne et al., 2011; Poulin et al., 2016; Abgottspon et al., 2015). These study designs allow for controlled conditions and replicable results. However, controlled experiments usually fail to cover the complexity of a real floodplain soil system (Ponting et al., 2020). Our study design did not involve temperature gradients, realistic hydrological flow conditions or intact soil structure. In this study, the artificial rainwater and the soil were equilibrated by shaking for a few minutes. However, the equilibration appeared to be incomplete with respect

to highly soluble chloride bearing minerals for the experiment with cornfield soil (Fig. S14). The incomplete equili-
bration is indicated by the temporal patterns of conservative ions ($Cl^-$, $K^+$ and $Na^+$) in soil solution (Figs. S7, S8) and
the difference in $Cl^-$ concentration between the soil solutions at $t = 6$ h and the same water-soil mixture shaken for 6
h (Fig. S14). These patterns are a result of a concentration gradient between supernatant water and the solution in the
soil pore space. They only became visible, due to high levels of conservative ions to start with, which most likely stem
from a fertilisation event prior to sampling the soil. Infiltration of supernatant water was facilitated by the sampling
of 4-6 % of the total added water at each time point. This resulted in a dilution of the soil solution. Consequently, the
continuous decrease in sulphate was not directly indicative for sulphate reduction, but the result of this dilution effect.
However, this effect did not directly affect the the release of soil bound elements (e.g. Hg, Mn, Fe, As) by e.g. reductive
dissolution (Figs. 2,3,4). It should also be noted that high initial $Cl^-$ concentrations in the soil solution, may influence
Hg solubility since $Cl^-$ is a complexant for $Hg^{2+}$ (Li et al., 2020) and this warrants further studies on the role of
inorganic fertilisation on Hg mobility.
**5. Conclusions**
We studied the effect of manure addition on the mobility of Hg in soil during a flooding-draining experiment. We
observed formation and size distribution changes of Hg colloids (ß-$HgS_{(s)}$, Hg-NOM) at environmental conditions in
soil solution by AF4–ICP–MS. The results of this study show that manure addition 1.) diminished HgT mobility, 2.)
facilitated Hg complexation with fresh NOM and formation of ß-$HgS_{(s)}$ and 3.) had only limited effect on net MeHg
production in polluted and periodically flooded soils.
Mercury was mobilized upon reductive dissolution of Mn oxyhydroxides in highly Hg polluted ($47.3 \pm 0.5$ mg $kg^{-1}$)
and NOM poor soils. The application of manure accelerated the release of Hg, facilitated the formation of colloidal
Hg and exhausted the mobile Hg pool within the first 7 days of flooding. This prevented Hg remobilization during the
second flooding period. Contrastingly, Hg was mainly released as particulate bound Hg in soils with moderate Hg
pollution ($2.4 \pm 0.3$ mg $kg^{-1}$) and high NOM levels. Ppresumably, due to its higher soil organic carbon content. This
relatively small pool of particulate Hg was exhausted within the first flooding period. In both soils, soil reduction
enhanced net MeHg production of a substantial part of the Hg pool as confirmed by MeHg formation upon flooding-
draining cycles. However, MeHg was either subsequently removed from the soil by advective transport of dissolved
MeHg in the soil column or transformed by reductive demethylation. We suggest that the temporal changes in net
MeHg production are limited by microbial activity of Hg methylators, given the similar net MeHg production in
treatments and soils with variable dissolved Hg levels. Microbial activity is likely to be stimulated by manure addition.
The release of Hg from polluted soils to downstream ecosystems does depend on both biogeochemical conditions as
well as on hydrological transport. Our experiment shows that redox oscillations (flooding-draining-flooding cycles)
of a polluted floodplain soil are likely to induce pulses of both Hg and MeHg to the downstream ecosystems. This is
supported with earlier studies (Poulin et al., 2016; Frohne et al., 2012; Hofacker et al., 2013). In contrast to NOM rich
soil systems, we show that the Mn dynamics may govern the release of Hg in highly polluted soil systems low in
NOM. Further, the application of additional NOM in form of manure facilitates soil reduction, contributed to the

transformation of Hg towards less mobile species reduced the Hg mobilization.However, effects of carbon amendments (organic amendments or biochar) are contrasting between enhancing (Li et al., 2019; Eckley et al., 2021) and diminishing (Beckers et al., 2019; Wang et al., 2020; Wang et al., 2021) Hg mobility. We therefor stress the need for characterisation of soil properties and especially NOM in future studies focusing on Hg mobility upon organic amendments (Li et al., 2019). We further emphasize the need of field trials integrating biogeochemical processes, hydrological transport and Hg soil-air exchange in order to establish Hg flux models to better understand *in situ* soil Hg mobility.

**Data availability.**

Details of analytical methods, AF4–ICP–MS fractograms are given in the Supplement. A complete dataset of the data used in this study is accessible at http://doi.org/10.5281/zenodo.4715110

**Acknowledgements.**

We acknowledge P. Neuhaus, J. Caplette, K. Trindade, K. Kavanagh, and D. Fischer for the help in the laboratory. We thank T. Erhardt at the Climate and Environmental Physics (CEP) at University of Bern for the ICP–TOF–MS analyses and Stephane Westermann at the Dienststelle für Umweltschutz (DUS) of the Canton Wallis for the help with site selection and sampling permissions. Soil temperatures have been provided by MeteoSwiss, the Swiss Federal Office of Meteorology and Climatology. Klaus Jarosch and Moritz Bigalke of the soil science group at the Institute of Geography at University of Bern gave valuable advice during the writing process.

**Author contribution.**

AM and LG designed the study. LG and AW preformed the incubation experiments. LG and IW performed laboratory analyses. LG and IW performed the data analysis. AM and VS supervised and financed the study. LG prepared the manuscript with contributions from all co-authors.

**Financial support.**

This work was funded the Swiss National Science Foundation (SNSF, Nr. 163661). VS and IW acknowledge the financial support of the SNSF R'Equip project Nr. 183292.

**Competing interests.**

The authors declare that they have no conflict of interest.

**Review statement.**

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

**Table 1: List of soil parameters for the two incubated soils (HMLC and LMHC) and manure (MNR). Uncertainties are given as 1σ standard deviation of triplicate experiments (method triplicates).**

| Parameter | Unit (dry.wt.) | Cornfield (HMLC) Concentration | SD | Pasture field (LMHC) Concentration | SD | Cow Manure (MNR) Concentration | SD |
|---|---|---|---|---|---|---|---|
| Land use | | Corn field | | Pasture | | - | |
| Depth | | 0 - 20 cm | | 0 - 20 cm | | - | |
| Soil Type (WRB) | | Fluvisol Gleyic | | Fluvisol Gleyic | | - | |
| spH | | 8.16 | | 7.84 | | - | |
| Water content | (wt. %) | 13.8 | | 8.5 | | 90.3 | |
| $C_{org}$ | wt. % | 1.92 | 0.01 | 3.45 | 0.01 | 45.22 | 0.09 |
| $N_{tot}$ | wt. % | 0.181 | 0.001 | 0.372 | 0.002 | 3.68 | 0.08 |
| $C_{org}/N_{tot}$ | - | 10.61 | - | 9.29 | - | - | - |
| S | g kg$^{-1}$ | 0.63 | 0.05 | 0.77 | 0.05 | 3.7 | 0.1 |
| Hg | mg kg$^{-1}$ | 47.3 | 0.5 | 2.4 | 0.3 | 0.045 | 0.001 |
| MeHg | µg/kg | 26.9 | 0.2 | 6.4 | 0.2 | <0.02 | - |
| MeHg/Hg | % | 0.06 | - | 0.28 | - | - | - |
| Al | wt. % | 0.91 | 0.05 | 1.05 | 0.04 | 0.0106 | 0.0003 |
| Fe | | 1.95 | 0.07 | 2.38 | 0.05 | 0.0336 | 0.0009 |
| Mg | | 1.25 | 0.07 | 1.39 | 0.05 | 0.49 | 0.03 |
| Mn | mg kg$^{-1}$ | 493 | 21 | 672 | 38 | 53 | 1 |
| P | | 1169 | 80 | 1044 | 85 | 8245 | 232 |
| Cr | | 56 | 4 | 64 | 5 | 0.68 | 0.01 |
| Co | | 10.75 | 0.06 | 11.22 | 0.43 | 0.4 | 0.2 |
| Ni | | 81.7 | 0.8 | 78.3 | 2.9 | 2.3 | 0.1 |
| Cu | | 40.1 | 1.2 | 28.0 | 0.7 | 13.1 | 0.6 |
| Zn | | 61.8 | 0.5 | 47.3 | 2.0 | 81 | 3 |
| As | | 11.74 | 0.07 | 16.04 | 0.72 | 0.8 | 0.4 |
| Cd | | 0.21 | 0.04 | 0.17 | 0.01 | 0.042 | 0.004 |
| Pb | | 20.8 | 0.5 | 18.34 | 0.5 | - | - |
| V | | 17.2 | 0.4 | 20.99 | 1.1 | 0.31 | 0.01 |
| Sr | | 137 | 2 | 202 | 6 | 45.9 | 1.6 |
| Cs | | 1.99 | 0.02 | 1.52 | 0.04 | - | - |
| Ba | | 60.2 | 1.1 | 76.9 | 1.6 | 9.1 | 0.5 |
| Ce | | 7.0 | 0.4 | 8.6 | 0.6 | - | - |
| Gd | | 0.94 | 0.03 | 1.00 | 0.05 | 0.021 | 0.001 |
| U | | 1.74 | 0.08 | 1.29 | 0.01 | 0.19 | 0.01 |
| Hg/Cu molar | ‰ | 366.3 | - | 25.73 | - | - | - |
| Hg/Mn molar | | 25.758 | - | 0.926 | - | - | - |
| Hg/$C_{org}$ molar | | 0.147 | - | 0.004 | - | - | - |
| Mn/$C_{org}$ molar | | 0.0056 | - | 0.0042 | - | - | - |






**Table 2: Description of the symbols and terms used for different filter fractions in the publication. The particulate fraction**
**is calculated as the difference of the 20 nm and the 10μm filtrate concentrations.**

| Filter Type | Filter size | Symbol (e.g. $HgT_x$) | Description |
|---|---|---|---|
| **Suction Cup** | 10 μm | $HgT_{<10\mu m}$ | Soil solution sampled directly from the suction cup contains a variety of particles (clay minerals, bacteria, Mn-/Fe-hydroxides, POM aggregates etc.). We refer to this fraction by adding the subscripts <10μm to the analyte symbol. |
| **Syringe Filter** | 0.02 μm | $HgT_{<0.02\mu m}$ | Soil solution <0.02μm is a cutoff size that may still carry colloids. We refer to this fraction by adding the subscripts <0.02μm to the analyte symbol. |
| - | - | P-HgT | Particulate Hg is calculated as: $PHg = Hg_{<10\mu m} - Hg_{<0.02\mu m}$ |
| - | - | P-HgT$_{rel.}$ | Relative particulate Hg is calculated as: $PHg_{rel.} = (Hg_{<10\mu m} - Hg_{<0.02\mu m})/Hg_{<10\mu m}$ |
| **AF4 membrane** | 1 kDa | $HgT_{<1kDa}$ | Molecules in solution under this cutoff size are not expected to have colloidal properties. Therefore, this range is referred to as "truly dissolved" in the text. |




**Table 3: Soil MeHg concentrations and net-methylation (MeHg/Hg) over the time of the experiment.**

| Treatment | day | n | Mean MeHg (µg kg⁻¹) | SD MeHg (µg kg⁻¹) | Range MeHg (µg kg⁻¹) | MeHg/Hg (‰) | net MeHg production (%) |
|---|---|---|---|---|---|---|---|
| HMLC | 0 | 1 | 26.9 | - | 26.9 - 26.9 | 0.57 | - |
| | 14 | 3 | 30.14 | 2.19 | 28.04 - 32.42 | 0.64 | 12.0 |
| | 28 | 3 | 52.04 | 10.65 | 39.74 - 58.25 | 1.1 | 73.1 |
| | 42 | 3 | 30.03 | 5.05 | 26.93 - 35.86 | 0.75 | -32.4 |
| HMLC +MNR | 0 | 1 | 26.9 | - | 26.9 - 26.9 | 0.57 | - |
| | 14 | 3 | 43.41 | 1.99 | 42 - 44.81 | 1.03 | 81.1 |
| | 28 | 3 | 57.79 | 13.79 | 41.88 - 66.41 | 1.24 | 20.7 |
| | 42 | 3 | 30.94 | 3.43 | 28.85 - 34.9 | 0.67 | -45.9 |
| LMHC | 0 | 1 | 6.4 | - | 6.4 - 6.4 | 2.72 | - |
| | 14 | 3 | 8.11 | 1.09 | 7.33 - 9.36 | 2.99 | 10.0 |
| | 28 | 3 | 12.07 | 1.1 | 10.81 - 12.87 | 4.11 | 37.2 |
| | 42 | 3 | 7.95 | 0.35 | 7.73 - 8.36 | 3.42 | -16.7 |
| LMHC +MNR | 0 | 1 | 6.4 | - | 6.4 - 6.4 | 2.69 | - |
| | 14 | 3 | 10.86 | 1.86 | 8.76 - 12.32 | 3.72 | 38.1 |
| | 28 | 3 | 14.31 | 0.17 | 14.12 - 14.43 | 4.7 | 26.6 |
| | 42 | 3 | 8.4 | 0.09 | 8.33 - 8.5 | 3.67 | -22.0 |





Figure 1 Schedule of preformed incubation experiment, samplings and measurements: Blue bars indicate soil flooding pe-
riods. Gray bars represent drained periods. The width of the columns is not proportional to the time of incubation. In the
treatments row the (≠) symbol indicates the addition of liquid manure to the microcosms specifically treated with manure
(+MNR). Triangles represent regular soil solution sampling points. Rectangles represent soil solution sampling for colloid
analyses. Diamonds represent time points for soil sampling. At -7 days, soil was sampled from the pooled soil directly before
the pre-incubation.

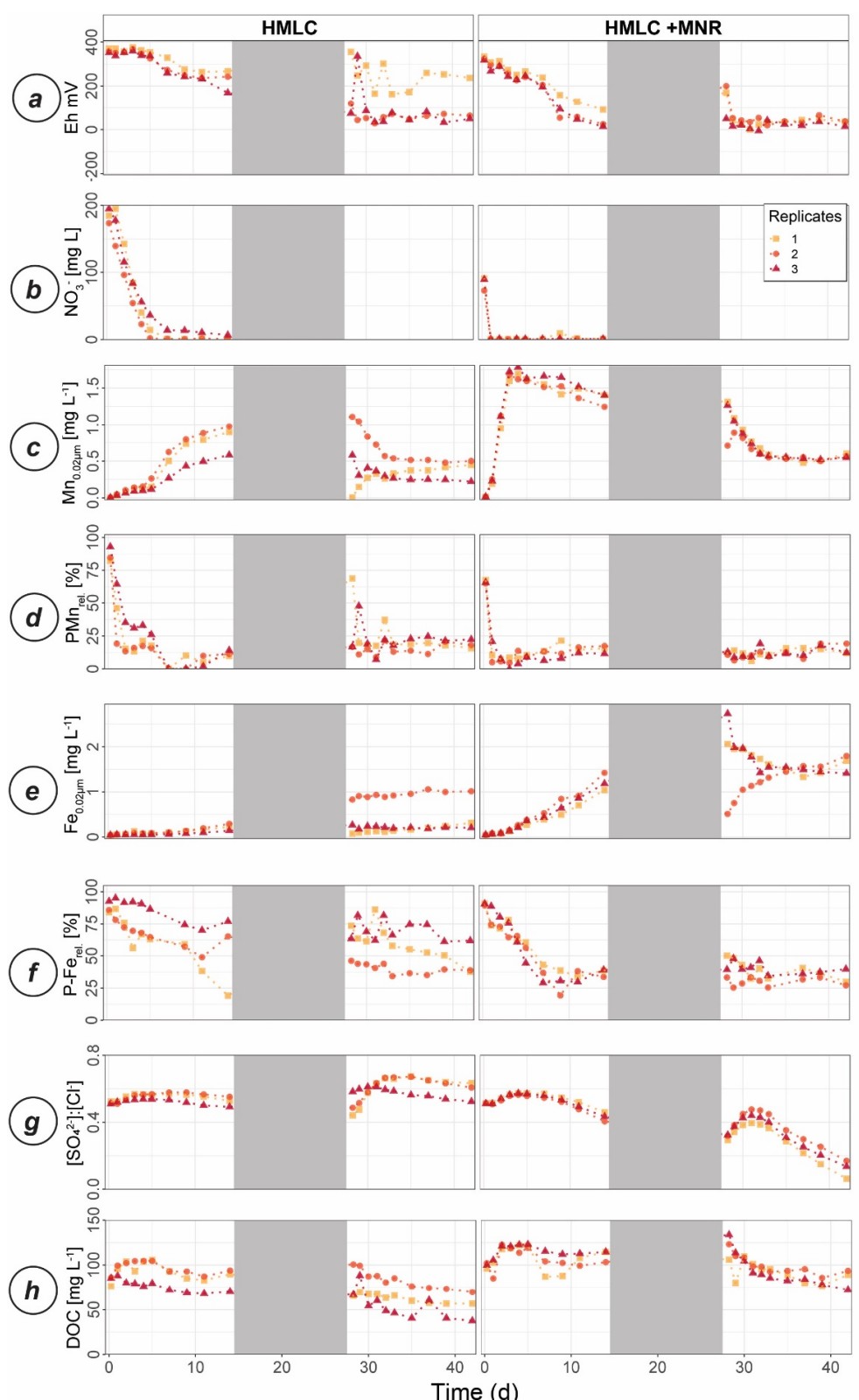


**Figure 2 Soil solution dynamics in cornfield soil (HMLC) incubations for redox potential (a),**
**redox reactive elements (Mn, PMn, Fe, P-Fe, $[SO_4^{2-}]:[Cl^-]$) (b-f) and dissolved organic carbon (h).**
**Lines between points were plotted to improve readability. The gray area indicates the drained**
**period.**

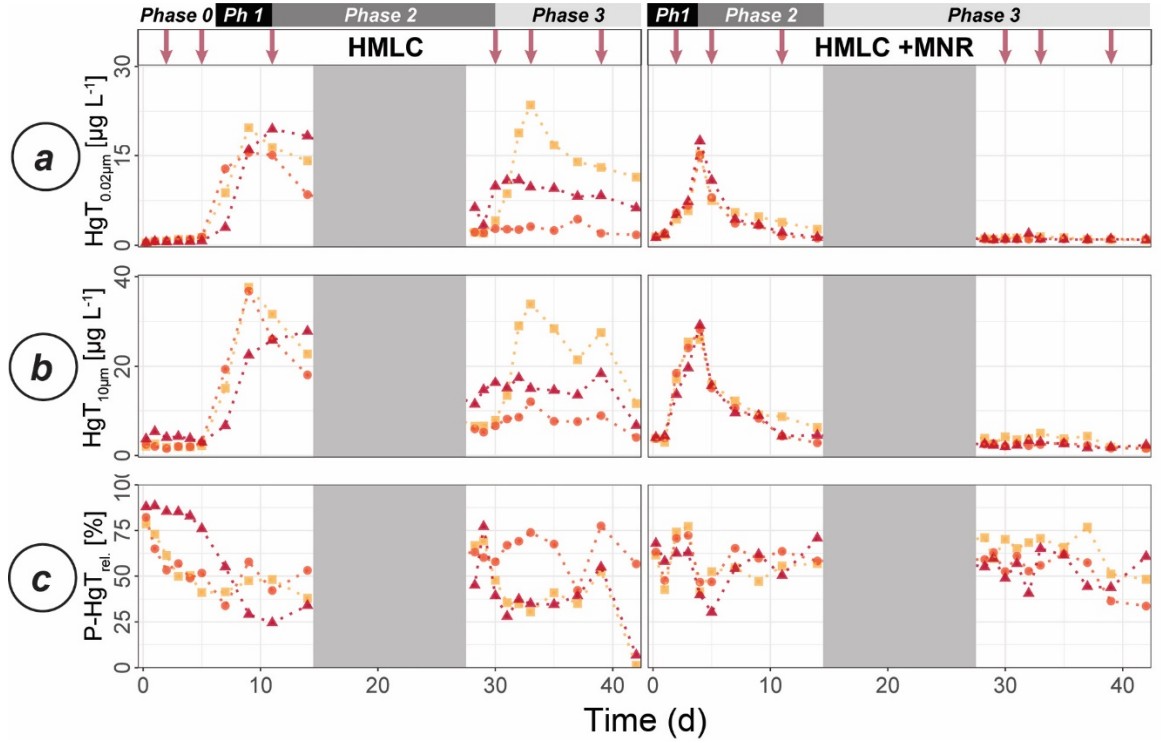


**Figure 3 Soil solution dynamics in cornfield soil (HMLC) incubations for Hg (a-c) subdivided in phases (0-3). Lines**
**between points were plotted to improve readability. The gray area indicates the drained period. Red arrows indi-**
**cate sampling days for AF4-ICP-MS analyses.**



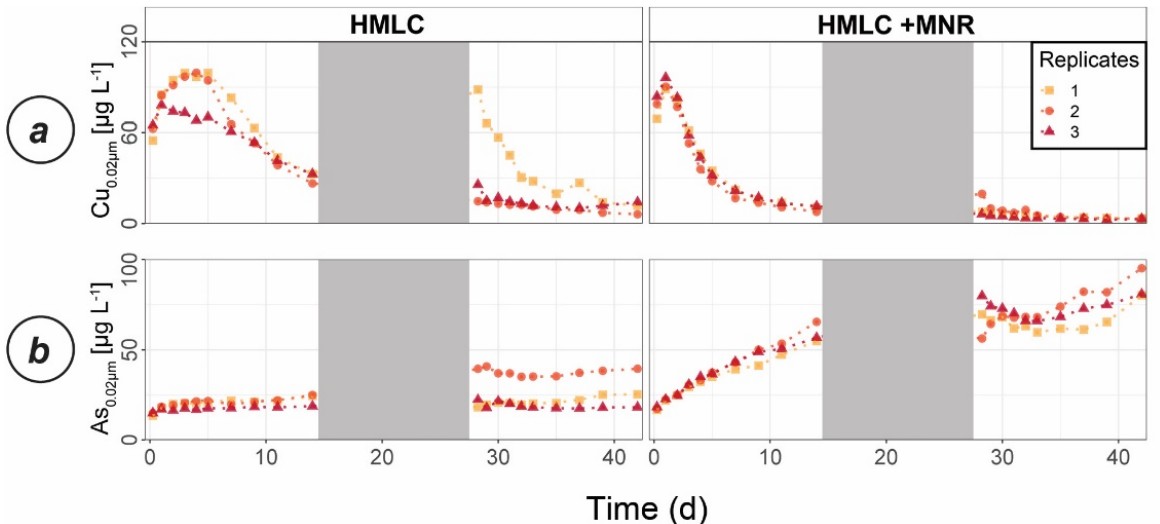


**Figure 4 Soil solution dynamics in cornfield soil (HMLC) incubations for Cu (a) and As (b). Lines between points**
**were plotted to improve readability. The gray area indicates the drained period.**



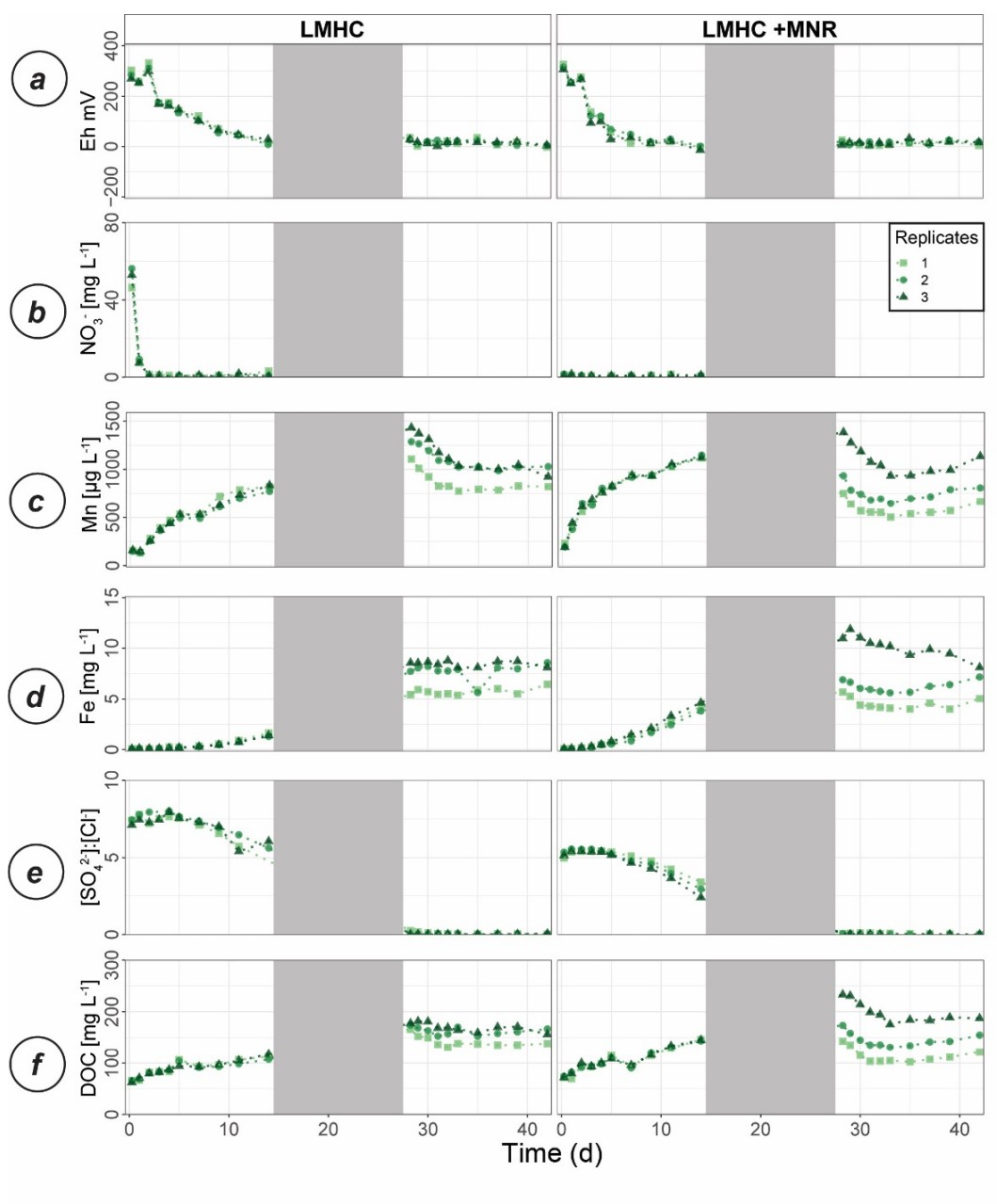


**Figure 5 Soil solution dynamics in pasture field soil (LMHC) incubations for redox potential (a), redox reactive elements (Mn, PMn, Fe, P-Fe, [SO$_4^{2-}$]:[Cl$^-$]) (b-f) and dissolved organic carbon (h). Lines between points were plotted to improve readability. The gray area indicates the drained period.**


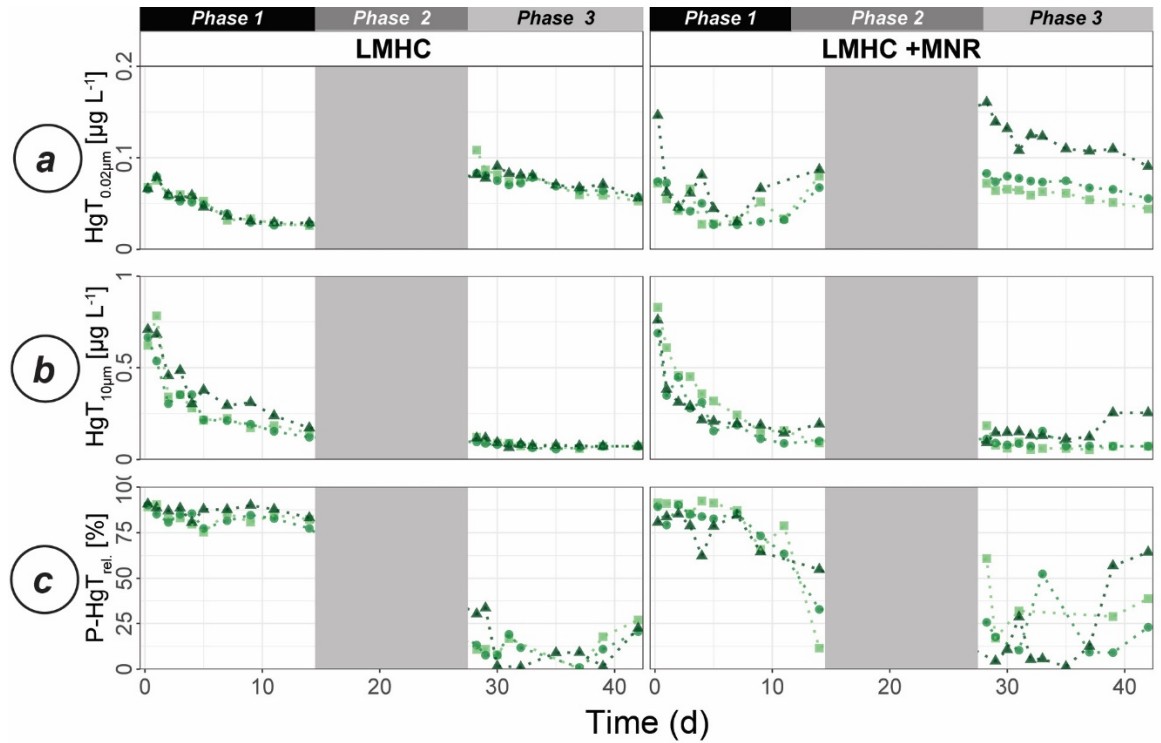


**Figure 6 Soil solution dynamics in pasture field soil (LMHC) incubations for Hg (a-c) subdivided in phases (1-3). Lines between points were plotted to improve readability. The gray area indicates the drained period.**



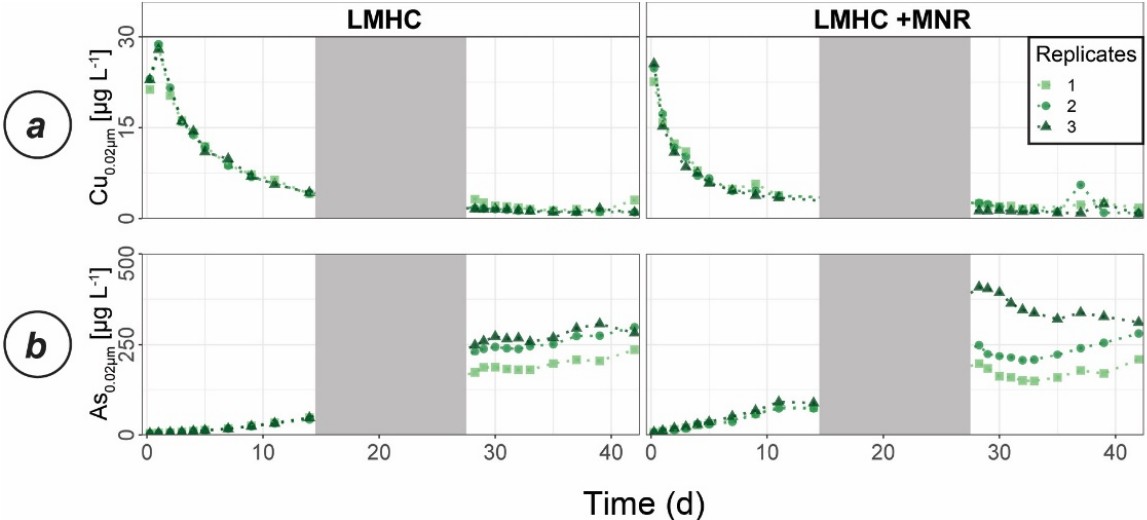


**Figure 7 Soil solution dynamics in pasture field soil (LMHC) incubations for Cu (a) and As (b). Lines between points were plotted to improve readability. The gray area indicates the drained period.**



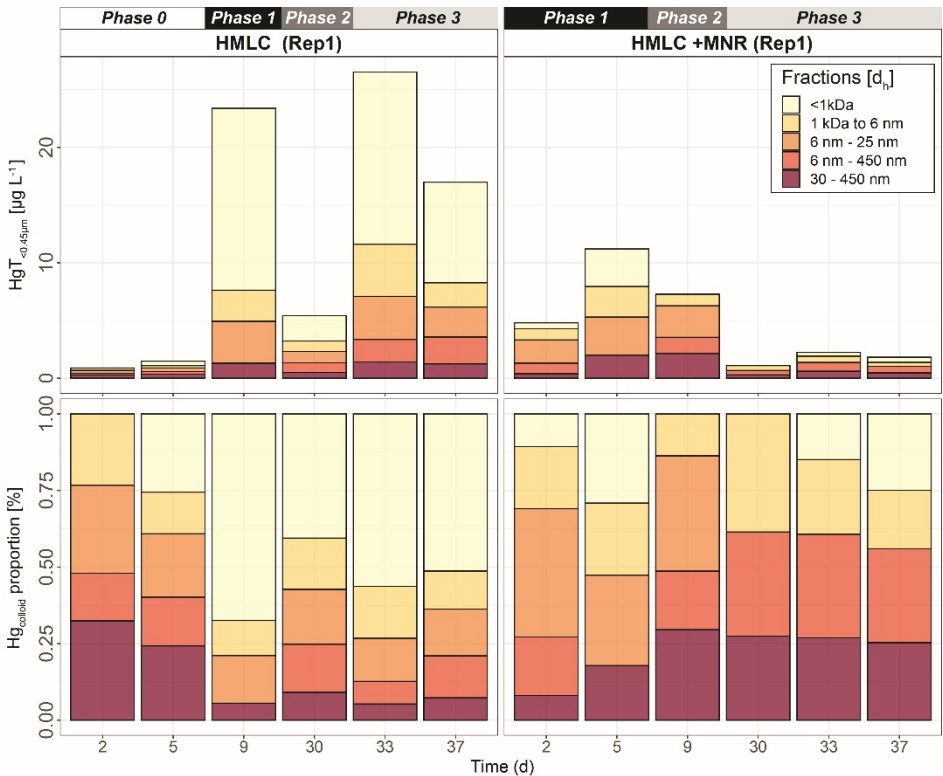

**Figure 8 Size distribution of Hg estimated after AF4 fractogram deconvolution for Rep1 of corn-field soil incubation (HMLC and HMLC +MNR) subdivided in phases (0-3). The concentration of HgT in size fractions was calculated using an external calibration of the ICP–MS directly after the AF4 run. The concentration of HgT in "< 1kDa" was calculated by subtracting the sum of the fractions from the HgT concentration in the same sample measured separately by ICP–MS. The fractograms of all analysed time points are shown in the supplement (Figs. S9-S12).**

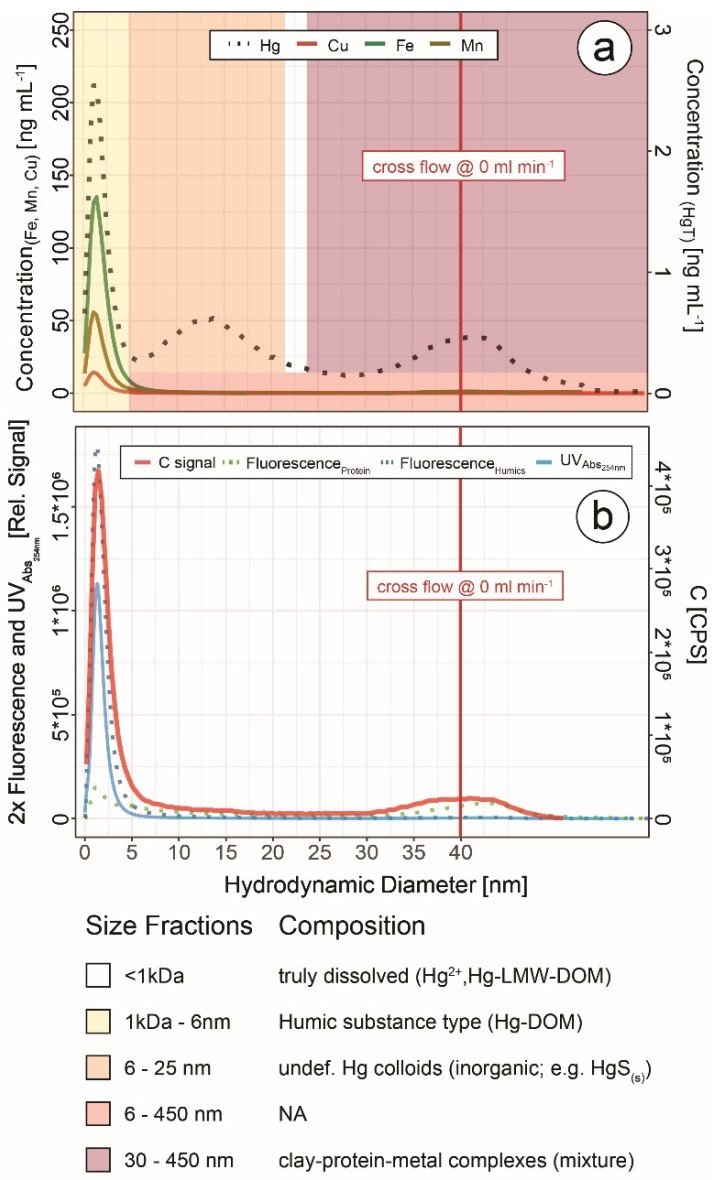

Size Fractions | Composition

| | <1kDa | truly dissolved ($Hg^{2+}$, Hg-LMW-DOM) |
| | 1kDa - 6nm | Humic substance type (Hg-DOM) |
| | 6 - 25 nm | undef. Hg colloids (inorganic; e.g. $HgS_{(s)}$) |
| | 6 - 450 nm | NA |
| | 30 - 450 nm | clay-protein-metal complexes (mixture) |


**Figure 9 Hg, Cu, Mn and Fe concentrations (a) and C signals (ICP–MS),**
**$UV_{254nm}$ absorbance and fluorescence signals (b) in colloids as a function**
**of hydrodynamic diameter (related to retention times on AF4) in a sam-**
**ple from HMLC at day 9 after flooding. These fractograms were ob-**
**tained at linearly decreasing crossflow from 2 to 0 mL $min^{-1}$ over 20 min.**
**The red line indicates the time point where the crossflow reached 0 ml**
**$min^{-1}$. Areas (yellow to red color) indicate size fraction ranges assigned**
**during deconvolution.**


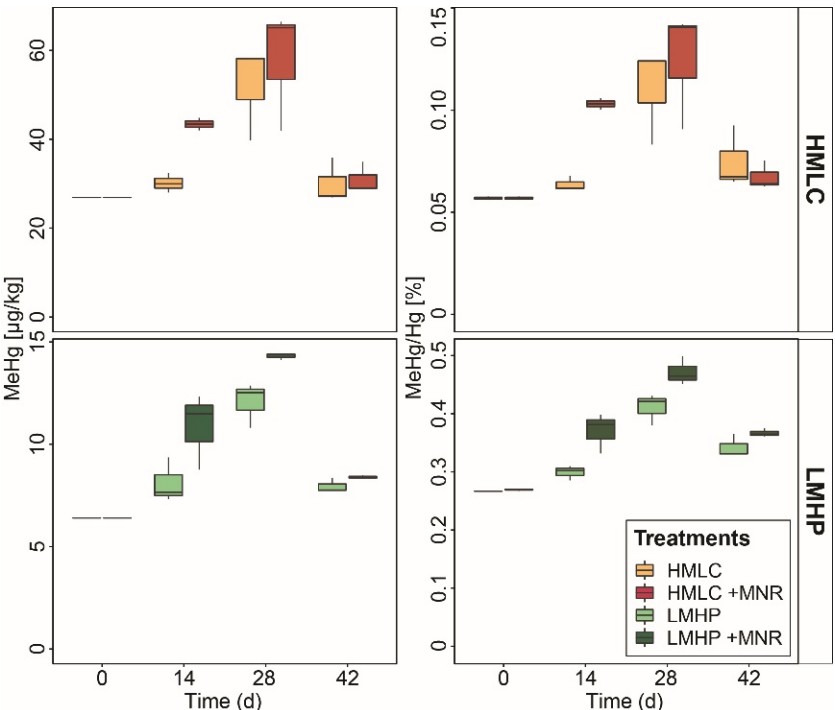

**Figure 10 Soil MeHg concentrations and MeHg/Hg ratios over the course of the experiment for corn field soils (HMLC, yellow/red) and pasture field soils (LMHC, lime/green). Highest net methylation was observed during first flooding for +MNR treatments and during the draining period for microcosms without manure addition. A significant decrease of MeHg/Hg was observed during the second flooding for all treatments.**

912