# Peer review of "Mercury mobility, colloid formation and methylation in a polluted fluvisol as affected by manure application and flooding-draining cycle."

_Biogeosciences, 2020_

## Author Comment (AC1)

**Response to:**

**Comment on bg-2020-466 – Jan Wiederhold**

*Referee Comment:*

This manuscript reports the results of laboratory experiments in which two Hg contaminated soil samples were incubated with and without addition of manure for six weeks over a controlled flooding-draining-flooding cycle. Soil solution samples were collected repeatedly during the experiments through open-pore suction cups, followed by measurements of colloidal and dissolved Hg, methyl-Hg and many other parameters using a variety of analytical techniques. The topic of the study is suitable for Biogeosciences and the results are novel and relevant for the large research community interested in mercury biogeochemistry.

The experiments were well-designed, the methods are described and validated in great detail, and the quality of the analytical data is high. The interesting results are presented and discussed in a detailed manner and supported by nice figures and tables (only with slightly too small font sizes for my taste in some cases). Some of the main findings include the association of Hg with Mn during the mobilization into soil solution after flooding, the lower Hg mobility in the manure-treated soils, a detailed characterization of Hg-bearing colloidal particles, the relatively large differences between the first and the second flooding period, and the inference that Hg methylation was limited by microbial activity/uptake rather than bioavailable Hg during the experiments. I congratulate the authors to their very interesting study and I recommend that the manuscript should be published in Biogeosciences after moderate revisions considering the following comments and suggestions.

My only general comment refers to the direct comparison of the results with previous related studies and therewith a more concise identification of the new insights generated by this study and their implications for Hg biogeochemistry in contaminated floodplains. I acknowledge that the authors present a thorough literature review in the introduction (a few additional recent studies are listed below), but the later parts of the discussion and conclusion sections could maybe still be improved by highlighting the similarities and differences of the new results with previous studies investigating contaminated soils from other field sites. Despite the detailed soil characterization and previous work at the site, the main binding form(s) of Hg in the soils at the studied contaminated site still remains somewhat unclear, making a direct transfer of the results to other contaminated legacy sites more difficult. Anyway, this is just an appeal to try carving out the specific new findings of the study and their implications to a larger extent than what is already done in the well-prepared manuscript. I look forward to seeing the final product in print.

*Author Response:*

We thank the referee very much for this positive review of our study. We acknowledge the need for comparison to similar studies and appreciate the suggestions made by the referee. With your valuable input, we were able to discuss our results more thoroughly.

*Referee Comment:*

l12: I would use "eco-systems" instead of "ecosystems".

*Author Response:*

Changed to "ecosystems".

***Referee Comment:***

l19: I am not sure whether the term "control soils" is helpful in this context. The same two soils were used in all experiments, once with manure addition and once without manure addition. The experiments without manure addition could be denoted as control experiments to assess the effect of the manure addition, but the soils are not "control soils" in my opinion.

***Author Response:***

We agree that the use of "control soils" can be misleading.

Changed L19 "control soils" to "soils" and removed the term "control" throughout the manuscript.

***Referee Comment:***

l21: I don´t think that you were able to monitor "methylation of Hg in the soil solution". You measured MeHg levels in the soil solution, but it´s not clear that the methylation process also took place in the soil solution.

***Author Response:***

Changed accordingly.

***Referee Comment:***

l22: "lower" instead of "lowest"?

***Author Response:***

Changed to lower.

***Referee Comment:***

l25-26: What do you mean by "proportional increase"? Do you refer to a higher fraction of colloidal Hg relative to total Hg in the manure vs. the non-manure experiments? Do the percent values indicate relative or absolute values? Maybe "higher relative" instead of "proportional"? Please rephrase to clarify.

***Author Response:***

We agree with the reviewer's comments: and changed accordingly to "a relative increase of colloidal DOM-Hg..."

***Referee Comment:***

l27: "Net Hg methylation" is not the same as "MeHg/Hg", but it could be maybe described as "increase of MeHg/Hg relative to the initial condition" if no absolute MeHg values can be compared.

*Author Response:*

We agree that "Net Hg methylation" can be misleading. In the new version we used the term "net MeHg production" when talking about changes in absolute MeHg concentrations. As these changes are the results of both methylation and demethylation processes in soil.

*Referee Comment:*

l47: Hg is not "found as FeS" but can be associated with this mineral phase. HgS could be both cinnabar or metacinnabar.

*Author Response:*

Changed from "found as" to "associated with".

*Referee Comment:*

l55: The term "immediate decrease" is not really clear in my view. A release of Hg into soil solution first causes a concentration increase. Maybe "relatively rapid" instead of "immediate"?

*Author Response:*

We agree with the reviewer's comment. The term "immediate" describes the closed possible sequential time relationship. "relatively rapid"/ "relatively fast" is a more suitable term to use for the described phenomenon.

Changed to: "relatively rapid".

*Referee Comment:*

l62: Here and throughout the manuscript: If a publication is cited with the author name in the text, then the year should be in brackets (here: "(2013)".

*Author Response:*

Agreed: Changed throughout the manuscript according to the format of BG. Here: Hofacker et al. (2013).

*Referee Comment:*

l72: I understand that Hg(II) binding to thiol-rich NOM is thermodynamically favored but I am not sure about the term "larger". Do you refer to molecular mass/size and can you give a reference to support this statement?

*Author Response:*

This is correct. Here we intent to refer to high molecular weight hydrophobic NOM which was show to have a high density of strong binding sites (thiols).

We changed this accordingly.

***Referee Comment:***

l89: "has" instead of "had"

***Author Response:***

changed to "has"

***Referee Comment:***

l96: The charge of sulfate is "2-".

***Author Response:***

Changed to "$SO_4^{2-}$"

***Referee Comment:***

l102: Weber et al. (2009) did not study Hg. Some additional Hg studies on temperate floodplain soils include for example Wallschläger et al. (1998, doi: 10.2134/jeq1998.00472425002700050009x) and Lazareva et al. (2019, doi: 10.1007/s12665-019-8253-9).

***Author Response:***

Corrected the misassignment of the reference used earlier in the text.

Removed "Weber et al. (2009)" added "Lazareva et al. (2019)"

***Referee Comment:***

l104: You may also refer here to the recent studies on Hg dynamics in similar experimental systems with biochar additions (e.g., Beckers et al., 2019, doi: 10.1016/j.scitotenv.2019.03.401 and 10.1016/j.envint.2019.03.040; Wang et al. 2020, doi: 10.1016/j.envpol.2020.115396 and 2021, doi:10.1016/j.chemosphere.2020.127794). Concerning similar experimental studies on other types of Hg-contaminated material, the recent studies by Zhu et al. (2018, doi: 10.1016/j.gca.2017.09.045) and Eckley et al. (2021, doi: 10.1016/j.envpol.2020.116369) could be of interest as well.

***Author Response:***

We appreciate the input the most recent literature and added a selection to this list to introduction and discussion.

Eckley et al. 2021, doi: 10.1016/j.envpol.2020.116369

Beckers et al., 2019, doi: 10.1016/j.scitotenv.2019.03.401

Wang et al. 2020, doi: 10.1016/j.envpol.2020.115396

Wang et al. 2021, doi:10.1016/j.chemosphere.2020.127794

*Referee Comment:*

l106: "studies" or "researchers" but not "researches"

*Author Response:*

Changed: "researches" to "researchers"

*Referee Comment:*

l110: Did you have an initial hypothesis on how the addition of manure would influence the system? If yes, it might be useful to present such a hypothesis here and then get back to it in the discussion/conclusion sections.

*Author Response:*

We added our working hypotheses, "Based on the presented state of knowledge, we hypothesize that the manure addition would accelerate the release of Hg by accelerated reductive dissolution of Mn-oxyhydroxides and eventually change the complexation of Hg in the system towards Hg-NOM complexes."

*Referee Comment:*

l118: Maybe better use the term "waste water releases" instead of "emissions" to clarify the pathway of the contamination. I think that many people primarily think about atmospheric pathways in the context of "emissions".

*Author Response:*

We acknowledge that in the scope of this study the expression "waste water releases" might be more precise.

Changed: "emissions" to "waste water releases"

To date, the pollution history is still not fully understood at this specific legacy site. Emissions of Hg(0) in the area can not be ruled out.

*Referee Comment:*

l118: The company did (and still does) not only produce acetaldehyde but also many other chemicals. Mercury was also used in several other processes including e.g., production of vinyl chloride and chlor-alkali electrolysis (see cited historical report by Glenz&Escher, 2011).

*Author Response:*

This is true. We referred to acetaldehyde because is still produced and was the main process applied during the time of highest Hg emissions (1960-1970s).

For completeness and clarity: removed "acetaldehyde producing" – added: "...chemical plant upstream historically using Hg in different processes (chlor-alkali electrolysis, acetaldehyde- and vinyl chloride production)"

*Referee Comment:*

l137: through

*Author Response:*

changed "though" to "through"

*Referee Comment:*

l138: Maybe better "Hg level" instead of "pollution". There could be also other pollutants present.

*Author Response:*

We changed "pollution" to "Hg level"

*Referee Comment:*

l147: add "and" after "soil"

*Author Response:*

For clarification we remove "soil" completely. Other possibility would be "soil's pore space".

"soil and pore space" may be misleading.

*Referee Comment:*

l180: I think that it should be "Table 2" instead of "Table 1" here (change numbers if this is mentioned first).

*Author Response:*

Changed numbers accordingly.

*Referee Comment:*

l185: In my opinion, there is no need to capitalize mineral names.

*Author Response:*

Changed: "Quartz, Albite, Orthoclase, Illite/Muskovite, Calcite." to "quartz, albite, orthoclase, illite/muskovite, calcite."

*Referee Comment:*

l191: I suggest adding the information which relative fraction of the total solution phase was withdrawn via sampling during the experiments. Could the lower water level already have had an influence on the results for the later sampling points?

*Author Response:*

Roughly, 4-6 % of the added water volume are sampled at each time point (30- 45ml). We added the information about the volume of water sampled in section 2.5 Soil solution sampling and analyses and added the sum of sampled porewater with respect to time in a supplement figure. We attached this figure at the end of this document.

Indeed, we think that sequential sampling from the bottom of a soil/water column may result in a relation between the sampling points. We now also further highlight this in the paragraph "Experimental Limitations". It is hard to evaluate if a correction of the data is reasonable.

*Referee Comment:*

l202: DOC concentrations are later reported as mg/L, so I suggest using the same unit here for the blank value.

*Author Response:*

Changed: molarity to concentration (w/v). "Incubation experiment blanks were below 4.75 mg L-1 and 22.4 µg L-1 for DOC and TNb, respectively."

*Referee Comment:*

l246: This section does not only describe Hg dynamics but also many additional parameters.

*Author Response:*

Changed: "Mercury dynamics (mobilization and sequestration)." To "Soil solution chemistry and Hg dynamics."

*Referee Comment:*

l251:  I assume 1SD of the triplicate experiments?

*Author Response:*

For clarification we added: "Uncertainties of soil solution parameters are display as 1SD of the triplicate incubation experiments throughout the manuscript." In section 2.5: Soil solution sampling and analyses.

*Referee Comment:*

l288: delete "but"?

*Author Response:*

We assume you refer to l280.

Change: Replaced "but" with "and" and added " the release" after "after".

*Referee Comment:*

l342: "suggests" instead of "suggest, "

*Author Response:*

Changed: "suggests" instead of "suggest, "

*Referee Comment:*

l343: There could be also other relevant Hg(II) binding sites in NOM even if all the thiol groups are saturated. Is there any indication in the literature that Hg(II) binding to Mn oxide phases would be preferred relative to, for example, Hg(II) binding to carboxyl groups in NOM or binding to Fe oxide phases? Anyway, I certainly agree that your interesting data suggests that Mn oxides play an important role for Hg cycling in the studied system.

*Author Response:*

Unfortunately, we could not find any literature on the competition between functional carboxyl-sites, Mn- and Fe-oxide surfaces. We acknowledge that input and would like to express the need of further research in the area.

*Referee Comment:*

l368: The spelling of "sulfate/sulphate" and "sulfide/sulphide" should be consistent.

*Author Response:*

Change: We chose the versions "sulphide and sulphate" and changed accordingly.

*Referee Comment:*

l386: Add "of " after "formation"

*Author Response:*

Change: Added "of " after "formation"

*Referee Comment:*

l415: Can you specify the approximate proportion of mobilized Hg relative to total soil Hg over the course of the experiment?

*Author Response:*

We added:

"...However, the released Hg-pool is relatively small compared the HgT levels of the soil. We estimate that about 12.8 ± 4.2 µg kg-1 Hg (0.02 % of $HgT_{soil}$) was evacuated by sampling during the experiment."

*Referee Comment:*

l439: words/values are missing after "up to"

*Author Response:*

Added the maximum value observed for MeHg in soil after the first flooding period.

44.81 µg kg-1

*Referee Comment:*

l455: delete "A"

*Author Response:*

Change: Accepted

*Referee Comment:*

l470: from

*Author Response:*

Change: Accepted

*Referee Comment:*

l473: Please explain how the sampling could have influenced the element concentrations in the remaining soil solution. As written before, I suggest describing the water level changes in the microcosm during the experiment and its potential effects on the investigated parameters.

*Author Response:*

Thank you for this suggestion. We further discussed the effect of the concentration gradient and the soil solution sampling influenced the element concentrations in the remaining soil solution in this section. We referred to the high relative amount of soil solution (4-6 %) sampled at each time point resulting in a change of water levels.

*Referee Comment:*

l477: Is chloride really an important component of inorganic fertilizers? I thought that most crop plants don´t like elevated chloride levels. And even though chloride forms could potentially form stable complexes with Hg(II) in soil solution, binding of Hg(II) to DOC (or generally NOM) is probably still dominant.

*Author Response:*

In Switzerland commercially used NPK fertilizers contain Potassium in the form of KCl and K2SO4. Fertilizers may contain up to 100% of the Potassium in the form of KCl. Please refer to https://www.google.com/url?sa=t&rct=j&q=&esrc=s&source=web&cd=&cad=rja&uact=8&ved=2ah UKEwibnPvqxrLvAhWLwAIHHROlDh8QFjAEegQIDhAD&url=http%3A%2F%2Fwww.landisense-oberland.ch%2Fmedia%2F0494af6b-2751-4de9-876a-3147f1180733%2FPQLbcw%2FMedien%2520LANDI%2520Sense%2520Oberland%2FTeaser%2FD%25 C3%25BCngersortiment%25202020.pdf&usg=AOvVaw00mIyoQoWZGrMndpatdBS5

a recent fertilizer catalogue.

As we show in Fig. S6 and S7 both K and Cl are in the range of 0.8 g/L and 0.5 g/L in soil solution at the beginning of the incubation. The ionic strength of this solution is close to brackish waters and surprisingly high. We assume that K and Cl concentrations would have decreased upon the next rain events in the area, given that K and Cl are conservative elements which are highly soluble and marginally interact with high specific surface minerals. This points towards a proximate fertilization of the soils in the area.

Although the binding of Hg to NOM might be favorable, an addition in of KCl to this extend in NOM poor soil might still influence Hg speciation in the upper 10 cm of the soil column.

We did not consider conducting aqueous geochemical modelling of Hg species. Here, the characterization of the DOM is crucial to get a good picture. We did not further characterize DOM.

*Referee Comment:*

l481: I suggest that you try specifying the observed "distinct effect" of the manure addition. You could potentially come back here to initially defined hypotheses (see comment above) and conclude whether you have verified or falsified them. I could imagine that such an approach might be helpful in further highlighting the novelty of the findings compared with previous work. This is a carefully conducted and well-described experimental study, but I believe that it might be possible to identify more clearly which specific insights on Hg cycling in contaminated soils were generated and how these findings could be relevant to other field sites and future work.

*Author Response:*

Thank you for this suggestion. We reformulated the conclusions, specified the effects of manure addition (e.g. formation of meta cinnabar, accelerated Mn oxyhydroxide reduction) and compared our work with previous studies to emphasize the novelty of our study.

*Referee Comment:*

l489: suggests

*Author Response:*

Changed: "suggest" to "suggests"

*Referee Comment:*

l489: Which changes in redox conditions do you refer to here? Higher/lower redox potential or do you mean that fluctuating redox conditions in general (irrespective of the direction) increase Hg methylation?

*Author Response:*

We refer to a soil reduction. Changed accordingly.

*Referee Comment:*

l490: Maybe better "is removed from the soil" instead of "declines from the soil"?

*Author Response:*

Change: "However, MeHg may subsequently either be removed from the soil by advective transport of dissolved MeHg in the soil column or be transformed by reductive demethylation."

*Referee Comment:*

l492: add "of" after "changes"

*Author Response:*

Change: Accepted

*Referee Comment:*

l492: Wording: Are the "temporal changes" really limited by "microbial activity"? Or rather "controlled by the extent of microbial activity"?

*Author Response:*

Changed

*Referee Comment:*

l493: Maybe "stimulated" instead of "facilitated"?

*Author Response:*

Replaced "facilitated" by "stimulated".

*Referee Comment:*

l497: It´s nice if your findings are supported by earlier studies, but I suggest highlighting the novelty of your findings (e.g., important role of Mn redox dynamics? decreased mobility due to manure addition? etc.).

*Author Response:*

The novelty of our findings was better characterized in the discussion and the conclusions in order to differentiate this study from previous ones.

*Referee Comment:*

l498: How does this finding compare with other studies in which organic amendments were added to Hg contaminated soils (see e.g., references listed above)?

*Author Response:*

Added comparisons with biochar amendments (Eckley et al. 2021, doi: 10.1016/j.envpol.2020.116369, Beckers et al., 2019, doi: 10.1016/j.scitotenv.2019.03.401, Wang et al. 2021, doi:10.1016/j.chemosphere.2020.127794) and organic amendments (Li et al. 2019 doi:10.1016/j.chemosphere.2019.05.234, 2019.)

*Referee Comment:*

l499-500: In my view, the sentence on "more work is needed" is superfluous. This is always the case.

*Author Response:*

Change Accepted

Changed: "We emphasize the need of field trials integrating biogeochemical processes, hydrological transport and Hg soil-air exchange in order to establish Hg flux models to better understand in situ soil Hg mobility."

*Referee Comment:*

l510-514: Please make sure that each sentence contains a verb.

*Author Response:*

Complete sentences.

*Referee Comment:*

l511: Stephane

*Author Response:*

Change: Accepted

*Referee Comment:*

l514: "advice" instead of "advises"

*Author Response:*

Change: Accepted

*Referee Comment:*

l582: Historische

*Author Response:*

Changed: Histoische to "Historische"

*Referee Comment:*

Figure 1: I suggest increasing the font size in the Table. This will be very small in a printed article.

*Author Response:*

Agreed. Changed font size. Moved table to landscape turned table to landscape.

*Referee Comment:*

Figure 2: This is a well-designed figure containing a lot of information. You could consider removing all the x-axes except the lowest one to make it a bit less busy. What about PFe (did you see a significant fraction of Fe colloids)? The "-1" in the y-axis caption of panel g should be superscript.

*Author Response:*

We followed the suggestion of the referee and:

1.) Created a new figure in response to reviewer 2

2.) Removed the x-axis labelling except the lowest one.

3.) We added P-Fe and discussed it in the main text.

4.) Made sure that all "-1" were superscript.

*Referee Comment:*

Figure 3: I suggest pointing out in the figure caption that Hg concentrations are shown here in ng/L instead of µg/L in Figure 2.

*Author Response:*

We followed the referee's suggestion:

1.) Mentioned the display of Hg concentrations in ng L-1

Figure 4: y-axis caption "colloid"

Changed.

Figure 5: y-axis caption "Fluorescence", legend "Composition" and "dissolved"

Changed.

*Referee Comment:*

Figure 6: I suggest that all y-axis ranges should start at zero to avoid a wrong impression of relative changes between the treatments. For the MeHg/Hg ratio, I suggest that you consistently use either percent or permil throughout the manuscript text and in figures and tables.

*Author Response:*

We followed the referee's suggestion and:

Adjusted the lower limit of the y-axis to 0 and changed MeHg/Hg ratios to the more frequently used %

*Referee Comment:*

Table 1: I suggest adding "Relative" before "Particulate" in the second last line.

*Author Response:*

Agreed with referee's suggestion.

*Referee Comment:*

Table 2: Please clarify the origin of the SD values (I assume based on triplicate experiments?).

*Author Response:*

This is correctly assumed by the referee.

Added: Uncertainties are given as 1σ standard deviation of triplicate experiments (method triplicates).

*Referee Comment:*

S3, l5: from

*Author Response:*

Change: Accepted

***Referee Comment:***

S3, l8: have

***Author Response:***

Change: Accepted

***Referee Comment:***

S3, l25: Merck

***Author Response:***

Change: Accepted

***Referee Comment:***

S3, l26: subscript "3"

***Author Response:***

Change: Accepted

***Referee Comment:***

S4, l25: define abbreviation DCM (dichloromethane)

***Author Response:***

Change: Accepted

***Referee Comment:***

S4, l29: add "to" after "transferred"

***Author Response:***

Change: Accepted

***Referee Comment:***

S9: I suggest clarifying in the figure caption that not only the map but also the high-resolution Hg concentration data was taken from the DUS report.

***Author Response:***

Change: Accepted

[Figure]

**Figure S1** The evolution of sampled solution. a.) and c.) display the sum of sampled solution during the incubation experiment for the HMLC and LMHC soil respectively. b.) and d.) display the relative volume of previously sampled solution with respect to added artificial rainwater. Blue lines mark the sum of water added during the experiment. The gray area indicates the drained period. The three shades of green/orange distinguish the 3 replicate incubators.

---

## Author Comment (AC2)

**Response to:**

**Comment on bg-2020-466 - Amrika Deonarine**

**Referee Comment:**

Very interesting data set on Hg-colloids and Hg methylation during flooding events! Agree with R1 comment on highlighting the novel and unexpected results in this study. Be careful though with overstating your conclusions, particularly with respect to HgS(s) formation (does the redox data support sulphate reduction/sulphide production?) and microbial activity (not measured)

**Author Response:**

We thank the A.D. for the interest in our study and appreciate the detailed comments. Please find the responses to you inputs.

*Referee Comment:*

Section 2.2: Was there a control for the manure only? AF4, Hg and MeHg data might be interesting for comparison.

*Author Response:*

We did not preform a microcosm experiment for manure only. However, Manure's Hg (45 ng g$^{-1}$) and MeHg (<0.02 ng g$^{-1}$) concentrations are given in Table 1. These levels are likely too low for a AF4 run. A MeHg and iHg spiking to the manure and a subsequent AF4 measurement would be indeed interesting and allow for comparison.

*Referee Comment:*

Line 144: Could you clarify what one application of the manure was?

*Author Response:*

We acknowledge the need of further clarification and added in the supplement:

"One application of liquid manure (0.6 % (w/w)) represents the recommended minimal application of 0.67 t km$^{-2}$ following the principles of fertilization of agricultural crops in Switzerland (Richner and Sinaj, 2017). This calculation assumes an affected soil depth of 10 cm and soil bulk density of 1.2 g cm$^{-3}$. This value is in the range of bulk density of soils from this area previously measured in our lab."

*Referee Comment:*

Line 241: There is a correction which can be made for the Fe contribution to SUVA. See

Poulin, B., et al. (2014). "Effects of iron on optical properties of dissolved organic matter." Environ. Sci. Technol. 48: 10098-10106.

*Author Response:*

We acknowledge the provided reference. However, the mentioned correction can not be applied.

The correction proposed by Poulin and co-authors is made for SUVA batch measurements. In our case, colloidal Fe can be both Fe(II) or Fe(III) (but not truly dissolved, as it is removed during the injection). We did not monitor Fe oxidation state of colloids during AF4 run and a the extrapolation of bulk Fe(II) and Fe(III) over the size spectrum would not be expedient.

**Referee Comment:**

Line 242: What exactly do you mean by "associated"? Which wavelengths were run for the humic-like fluorophores? Why was FLD run – what does it provide in addition to SUVA?

**Author Response:**

We have removed "associated" and replaced by "co-eluted". Fluorescence is more specific for online detection of humic substances-like components at an Excitation WL = 270 nm and an Emission WL = 460 nm, and allow to distinguish from protein-like components (Excitation WL = 280 nm and an Emission WL = 350 nm). Using AF4-FLD-UVD most of the studies agreed with lower size-range distribution of fluorogenic components compared to absorbing components, certainly due to the overlap absorbance of slightly higher sized iron-clusters or small ion colloids.

**Referee Comment:**

Line 320: Why do you think this fraction consists of HgS colloids?

**Author Response:**

In this paragraph we do not discuss colloid fractions. We assume the comment refers to Line 302. As stated in the text the Hg colloid fraction is neither overlap with NOM nor signals of the ICP-MS (e.g. Fe, Mn, Cu). Therefore, we hypothesise that this signal might originate from a HgS species (likely nanoparticulate meta cinnabar). The relative increase of this fraction was observed at the onset of sulphate reduction in the HMLC +MRN MC.

**Referee Comment:**

Line 322: Consider including values here.

**Author Response:**

Since the values can be found in the Table 3, we did not think it necessary to repeat them here.

**Referee Comment:**

Line 340: Red-S and Hg concentrations should be expressed in mol/g. Also, Hg can complex with other functional groups such as O-containing functional groups in OM. How does this fit into the competition scenario between OM and Mn oxides?

Is it possible considering the pH and pzc of Mn oxides that Hg can adsorb to the surface? Are there any other studies which have reported Hg- Mn interactions?

*Author Response:*

Thank you for this interesting input. Early lab experiments report the adsorption capacity for $Hg^{2+}$ on MnO2 surfaces to be 15 mmol kg$^{-1}$ over a wide range of pH (5-11)(P. Thanabalasingam and W. Pickering, 1985). This is in accord with the experimentally assessed pzc of $MnO_2$ of (4.9-4.1) (Miyittah et al., 2016). Considering these references, the presence of 493 ± 21 mg kg$^{-1}$ Mn in the HMLC soil and an oversaturation of strong binding sites of SOM in our soil, sorption of Hg on MnO2 is likely.

*Referee Comment:*

Line 357: Have you considered that OM can directly reduce Mn oxides or act as an electron shuttle?

*Author Response:*

Direct reduction of Mn oxides is a possible mechanism and was added here.

*Referee Comment:*

Line 367: There are many minerals which form black precipitates. Is there geochemical modeling data or XRD data to support this? Does the redox data support sulphide production?

*Author Response:*

Unfortunately, we could neither consider geochemical modelling nor XRD as suitable to identify the precipitates due to the following reasons.

1.)      XRD is a very powerful technique for characterisation of crystal structures of solid materials. However, characterisation of nano particles and solids with low crystallinity require a high sample purity (consider: https://doi.org/10.1021/acsnano.9b05157). This fact makes is rather difficult to characterize the newly formed phases (likely semi-crystalline phases) by XRD.
We analysed bulk soils before and after the experiment using XRD. Like this it was only possible to characterize the major crystal phases of the soil matrix (see manuscript: section 2.4).
In our opinion, the analyses of the black precipitates in question would require a very sensitive and time-intensive purification procedure when using our available XRD techniques and was beyond the scope of the study.

2.)      Further, we did not consider conducting aqueous geochemical modelling of Hg species. Here, the characterization of the DOM is crucial to get a good picture. Unfortunately, we did not further characterize DOM.

Concerning the redox data:
In incubators where manure was added we observed a vast decrease in Eh. It ranged between 100 and 0 mV at the time of the formation of the "black precipitates". Earlier incubation studies observed the onset of sulphate reduction already a 0mV. It is assumed that the bulk Eh measurements of the sampled pore water do not entirely reflect the redox conditions in the different pores and aggregates of the soil microcosm. Further, we used a flow-through system to measure Eh of the soil solution and oxidation of the sampled water between Microcosm and Eh probe cannot be out ruled although highly unlikely since the device is specifically made for this

purpose. Despite the rather high Eh measurements, we are convinced sulfate reduction and meta cinnabar precipitation is taking place. Now, we introduced $[SO_4^{2-}]:[Cl^-]$ to monitor sulphate reduction. Sulphate concentrations were not directly indicative of the onset of sulphate reduction. This is due to a chemical gradient between supernatant rainwater and soils solution demonstrated by the continuous decrease in concentration of conservative ions ($Cl^-$, $Na^+$, $K^+$) (Sect. 4.4).

*Referee Comment:*

Line 368: Is there a reference for the formation of sulphide minerals in meso- and micropores?

*Author Response:*

With this line we wanted to emphasize that redox potential is not an exact measure for ongoing redox processes. Eh measurement may only be used qualitatively in complex soil systems.

*Referee Comment:*

Line 386: Does the redox data support sulphide production?

*Author Response:*

Despite the rather high Eh measurements, we are convinced sulphate reduction and meta cinnabar precipitation is taking place. Now, we introduced $[SO_4^{2-}]:[Cl^-]$ to monitor sulphate reduction. Sulphate concentrations were not directly indicative of the onset of sulphate reduction. This is due to a chemical gradient between supernatant rainwater and soils solution demonstrated by the continuous decrease in concentration of conservative ions ($Cl^-$, $Na^+$, $K^+$) (Sect. 4.4).

See comments on L367

*Referee Comment:*

Line 406-408: This is very interesting. How do you think this relates to the decrease in Hg-0.02 um after 4 days for HMLC+MNR?

*Author Response:*

In case of HMLC+MNR, the Hg colloidal distribution was dominated by the presence of larger fractions (30 - 450 nm). Manure addition facilitates the soil reduction process and is a source of POM and larger NOM aggregates. As an effect of manure addition, Hg adsorb to larger NOM-aggreagtes and/or form nano particulate meta-cinnabar.

*Referee Comment:*

Line 404: Complexation is driven by thermodynamics and not necessarily by ligand concentration. Geochem modelling might help support this statement on chloride complexation.

*Author Response:*

Cl is present at extremely high concentrations. As we show in Fig. S6 and S7 both K and Cl are in the range of 0.8 g $L^{-1}$ and 0.5 g $L^{-1}$ in soil solution at the beginning of the incubation. The ionic strength of this solution is (surprisingly) close to brackish waters. We assume that K and Cl concentrations would have decreased upon the next rain events in the area given that K and Cl are conservative elements which are highly soluble and marginally interact with high specific surface minerals. This points towards an inorganic fertilizer application before the field sampling.

Although the binding of Hg to NOM might be favorable, an addition of such high amounts of KCl in NOM poor soil might still influence Hg speciation in the upper most cm of the soil column.

We could not consider conducting aqueous geochemical modelling of Hg species because the characterization of the DOM is crucial to get a good picture ande could not further characterize DOM.

*Referee Comment:*

Line 415: What is the fraction of the total Hg in the "small pool"?

*Author Response:*

Unfortunately, we did not use methods to analyse the solid speciation of our soil (sequential extraction procedures, or EXAFS) However, we estimated that we sampled approximately (0.02 % of the soils HgT) during the whole experiment. Therefore, the proportion of Hg mobilized from soil matrix seams relatively low, compared to its total content.

*Referee Comment:*

Line 425: Does the redox data support sulphide production?

*Author Response:*

See answer to the comment above, on L367

*Referee Comment:*

Line 442: What were the concentrations of bioavailable Hg?

*Author Response:*

Thank you for this question. We agree that bioavailability is more complex than a chemical measurement, no total chemical measurement can account for bioavailability. Some tools available can relate potential bioavailability, like DGT measurements (Ndu et al., 2018). Following this it is expected that higher truly dissolved Hg could be more bioavailable than other size fractions. However, the study also showed that filter passing <0.2 µm filter passing Hg might not be a good measure for bioavailability.

We included this in the discussion about bioavailable concentration and did not give concentrations, as this would be too farfetched.

*Referee Comment:*

Figure 3: Is there Fe data?

*Author Response:*

We added concentrations of $[SO_4^{2-}]:[Cl^-]$ and Fe to the Figures.

A complete dataset of this study is accessible in the data repository ZENODO http://doi.org/10.5281/zenodo.4058676 Fe data is also included there as well.

*Referee Comment:*

Figure 4: Consider revising the legend labels to be more descriptive of the different fractions. There is overlap between the 6-25, 6-450 and 30-450 nm size fractions, which makes interpreting the Hg proportion data difficult (sum to 100%).

*Author Response:*

The deconvolution of the fractograms included an intermediate fraction of Hg bearing colloids ranging between $d_h = 6$ nm and $d_h = 450$ nm depending on the sample. This fraction was added to refine the fractogram fittings and indicates that this population overlap a more polydisperse Hg particle population. The sum of colloid fractions was in accord with the total Hg in the filter fraction of 0.45µm filtrates.

*Referee Comment:*

Table 3: Be careful with de/methylation. What you are quantifying is a decrease in net methylation and not necessarily demethylation processes.

*Author Response:*

This is right. In the new version we consider this remark. We replaced methylation and demethylation with "Net MeHg production" when discussing our results.

**References**

Miyittah, M. K., Tsyawo, F. W., Kumah, K. K., Stanley, C. D., and Rechcigl, J. E.: Suitability of Two Methods for Determination of Point of Zero Charge (PZC) of Adsorbents in Soils, Communications in Soil Science and Plant Analysis, 47, 101–111, doi:10.1080/00103624.2015.1108434, 2016.

Ndu, U., Christensen, G. A., Rivera, N. A., Gionfriddo, C. M., Deshusses, M. A., Elias, D. A., and Hsu-Kim, H.: Quantification of Mercury Bioavailability for Methylation Using Diffusive Gradient in Thin-Film Samplers, Environmental science & technology, 52, 8521–8529, doi:10.1021/acs.est.8b00647, 2018.

P. Thanabalasingam and W. Pickering: Sorption of mercury(II) by manganese(IV) oxide, Environmental Pollution Series B, Chemical and Physical, 10, 115–128, 1985.

---

## Author Comment (AC3)

**Response to:**

**Comment on bg-2020-466 – Brett Poulin**

*Referee Comment:*

The study titled "Mercury mobility, colloid formation and methylation in a polluted fluvisol as affected by manure application and flooding draining cycle" aims to identify the release dynamics of Hg in two soils under two conditions (with and without manure) over two flooding periods. Two soils were characterized and incubated in laboratory microcosms with synthetic rainwater and with/without manure over two flooding cycles. Pore water was documented at numerous points over the two flooding periods, and measured for total Hg, metals, anions, cations, DOC, and pH and Eh. Colloids were collected at 3 time points during each of the 2 flooding periods, and AF4 measurements determine the size distribution and some elemental composition (Hg, Cu, Fe, Mn, carbon). Methylmercury was quantified in the soil a 4 time points between t=0 and t=final conditions.

Overall, the study documents some nice results from the incubation experiment that test the effects of soil properties and manure addition. The study design and methods are well done, and I agree with the majority of the conclusions. However, my main comments are about the presentation of the work and ways to improve the clarity in presentation. I have itemized general comments and specific comments that should be addressed by the authors before considering this work for publication. The authors are encouraged to edit the manuscript thoroughly for editorial clarity. I did not identify all the sentences and statements that were unclear, but have listed some editorial comments in the Specific Comments section below.

*Author Response:*

Thank you very much for the constructive and thorough comments on our study. These greatly contribute to the improvement of the manuscript and we edited it accordingly to better clarify the presentation of the results and outcomes of our study.

*Referee Comment:*

The importance of sulfate reduction should be revisited in this paper, as inorganic sulfide will scavenge pore water Hg(II) and result in authigenic formation of β-HgS. There is very little to no discussion of the decrease in sulfate concentrations in the microcosms, which indicates sulfate reduction and is a key biogeochemical transformation that can result in Hg partitioning back to the soils. Figure S7 in the SI shows very high levels of sulfate at the start of the experiments (150-1000 mg/L) and drastic decreases in concentration with flooding time.

*Author Response:*

We agree with the reviewer and have now thoroughly discussed the importance of sulphate reduction in the manuscript. However, due to the limitations described in the manuscript (gradient between artificial rainwater on the surface and soil solution, as seen for $Cl^-$, $K^+$ and $Na^+$), we are introducing the use of the molar ratios of $SO_4^{2-}$ to $Cl^-$ ($[SO_4^{2-}]:[Cl^-]$)to monitor sulphate reduction. During sulphate reduction, sulphate to chloride ratios will decrease as in geochemical terms $Cl^-$ is a conservative species.

*Referee Comment:*

For the presentation of the Microcosm results, and figure presentation, I recommend the authors (1) use the redox ladder to guide the initial presentation, (2) consider discussing the Hg release dynamics in terms of "stages" or periods of time describing trends in the concentrations, and (3) detail the release dynamics of the other metals separately. Regarding item 1 of the redox active elements, in Lines 247-255, there is no mention of Fe or sulfate and all pertinent constituents (nitrate, Mn, Fe, sulfate) should be presented together in a single figure (at present, the reader has to look to the SI and main text figures). The observation that reductive dissolution of Fe wasn't observed in Flooding period 1 is still a result that needed to be stated, and there is no mention of the decrease in sulfate from ~1000mg/L to 500 mg/L in flooding period 1 of the HMLC incubations. Regarding item #2 of the Hg release dynamics, on Lines 274-278, you may consider revising to describe the release dynamics in 'stages'. "Concentrations of Hg were low between X-X days (phase 1), increased to a maximum at 4 days (phase 2), and decreased between 4 and 14 days (Phase 3)." These same 'stages' could be references when describing the colloidal data. Regarding the third item on other metal contaminants, the study presents data on diverse metals (Cu, and all metals in Figure S8), but Cu is the only metal discussed. The authors need to discuss the data they present in all figures, otherwise it is unclear why those data are presented in the first place. I commend the authors for a nice study and recognize that presenting the various non-metal metal data is challenging.

*Author Response:*

We thank the referee for these constructive suggestions. Regarding item 1.) We drew a figure including $NO_3^-$, Mn, Fe and $[SO_4^{2-}]:[Cl^-]$ together with DOC in on figure and discussed soil reduction, including Fe reduction and sulphate reduction. Due to the already mentioned experimental limitations we used $[SO_4^{2-}]:[Cl^-]$ to monitor sulphate reduction in the experiment. Regarding item 2.) We drew a new figure including HgT concentrations for the different filtrates and introduced Phases as suggested. These Phases were also subsequently used in the discussion of the Hg colloids. Regrading item 3.) To introduce and discuss the dynamics of all measured trace metals in our experiment would be beyond the scope of this manuscript. However, the importance of metallic Cu was demonstrated earlier in soils with high Cu:Hg ratios (https://pubs.acs.org/doi/10.1021/es4010976). We acknowledge the referees view that what is displayed should be discussed and excluded parameters which were not discussed in the manuscript.

As suggested, the new figures are highlight 1.) redox processes, 2.) Hg measurements, 3.) trace elements. They are attached at the end of this document.

*Referee Comment:*

Have the authors considered including an analysis that estimates if the thiol content of the DOM was exceeded in their experiments, to contrast with the soil analysis (Lines 339-345)? The strong binding site capacity of DOM has been quantified to be ~5 nmol/mg DOM (https://pubs.acs.org/doi/10.1021/es025699i). They can assume DOC is 50% of the DOM, and compare the strong binding site capacity of the DOM to the total Hg concentration < 0.02 um.

Further, they could estimate how the addition of manure changed the Hg binding state (saturated vs unsaturated) in both the soil and pore water.

*Author Response:*

We did not measure thiol content of the DOM. However, this would be a good way to better characterize the DOM in future experiments.

We used the suggested factors to calculate the saturation of Hg and included this into the discussion. However, we did not conduct a geochemical modelling where we included aqueous $Cl^-$ and sulphide species. Therefore, these estimates have to be taken with caution.

*Referee Comment:*

As a reader, it will improve clarity if you spell out the various soils and treatments. I had to go back repeatedly to the methods to remind myself what the various acronyms meant ("HMLC"). This is important because of the two soils and two treatments (control vs manure).

*Author Response:*

We acknowledge and share the referees view, that abbreviations are not an optimal way to discuss the data. However, we think that for displaying and presenting the data in Figures, Methods and Results this practice is more accessible.

*Referee Comment:*

Line 28 – consider deleting "again".

*Author Response:*

Accepted.

*Referee Comment:*

Line 30 – consider "formation and aggregation" of …

*Author Response:*

Changed to "formation and aggregation"

*Referee Comment:*

Line 46-47 – There is "cinnabar" in the environment, but in the form of mineral deposits or associated with mercury mining activities. In the vast majority of environments, mercuric sulfide is present as authigenic nanoparticulate metacinnabar (β-HgS nano).

*Author Response:*

Changed to "..associated with FeS(s) or found as cinnabar (HgS(s)) or meta-cinnabar (ß-HgS(s))." For clarification.

*Referee Comment:*

Line 59 – consider deleting "e.g.".

*Author Response:*

Accepted

*Referee Comment:*

Line 62-63 – consider expanding to include the microbial process. At present, it reads awkward because Hofacker 2013 and 2015 are referenced but it is somewhat unclear what the 2015 study contributed.

*Author Response:*

Accepted. Added: "[…]which were shown to be formed by bacteria (*Clostridium* species) (Hofacker et al., 2015)"

*Referee Comment:*

Line 66 – The first reports of DOM facilitating the dissolution of HgS were by https://pubs.acs.org/doi/10.1021/es9804058. This study should be cited.

*Author Response:*

Accepted and added.

*Referee Comment:*

Line 66-68; here you reference "altering the reaction kinetics of HgS(s) formation", in which case you should cite Ravichandran et al., 1999 and Deonarine and Hsu-Kim, 2009.

*Author Response:*

Accepted and added

*Referee Comment:*

Line 72-74; Ligand exchange is important, but in the vast majority of systems there is an excess of strong thiols binding sites in the DOM for all the Hg(II). This was first documented by https://pubs.acs.org/doi/10.1021/es025699i and should be integrated into this sentence.

*Author Response:*

Haitzer et al. 2002 suggest that the lipophile fraction of DOM sampled from an aquatic system contain high amount excess of thiols sites. We integrated this reference in this paragraph and added a sentence about high density of strong binding sites in high molecular weight NOM.

*Referee Comment:*

Lines 66-70 and 430-431; the authors need to cite primary literature that document how DOM controls the nanocrystalline structure of β-HgS particles, which is a key property influencing the bioavailability of mercury under sulfidic conditions. Two paper that should be considered due to conditions that closely represent natural conditions include https://pubs.acs.org/doi/abs/10.1021/es201837h and https://pubs.acs.org/doi/10.1021/acs.est.7b02687.

*Author Response:*

We considered the two articles which report experimental evidence of meta-cinnabar formation close to our conditions. This and the decrease in $[SO_4^{2-}][Cl^-]$ further supported our hypotheses of HgS formation in the manure added runs.

*Referee Comment:*

Line 88; the authors may consider also looking at a recent paper on OM amendments to mine tailings. https://www.sciencedirect.com/science/article/pii/S0269749120370585

*Author Response:*

Thank you for this input we Cited this paper in the introduction and added it in the discussion.

*Referee Comment:*

Line 92-93; The authors are encouraged to revise this sentence, as it could be improved to highlight the various environments where methylation is prominent (riparian zones, saturated soils etc) because of the redox conditions.

*Author Response:*

Accepted. Changed to: "Environments of elevated Hg methylation (riparian zone, estuary) are also places of elevated NOM degradation and mineralization due to temporal changes in redox conditions."

*Referee Comment:*

Lines 108-2213; somewhere in this paragraph it should be mentioned that "microcosm experiments" were carried out.

*Author Response:*

Accepted.

*Referee Comment:*

Line 112 – revised to "0.02 and 10 µm" or an equivalent term. At present, "0.02/10 µm" is a fraction and doesn't make sense to me.

*Author Response:*

We agree, this may be confusing. Changed to "0.02 and 10 µm"

*Referee Comment:*

Line 132: First, this sub-header should read "Microcosm Experiments". It is confusing to call them "incubations" when later you refer to them as microcosms – please be consistent and in all instances state "microcosms". Second, in a section below you detail the "Incubation experiment blanks" but those are not detailed in this section, and they should be.

*Author Response:*

Changed. First, we renamed the section added the details ask for in the section "2.2 Microcosm Experiments". Second, we added information about blanks in "2.5 Soil solution sampling and analyses". "[...]Incubation experiment blanks were taken by sampling MilliQ water through from an empty 1 L borosilicate aspirator bottle 3 times throughout the experiment.[...]".

*Referee Comment:*

Line 135-136 – revised to "After the initial incubation period soils were used in the flooding and draining experiments, which were conducted in 1 L borosilicate glass aspirator bottles (Fig. S2)." It would appear Fig. S2 should be called out.

*Author Response:*

Accepted and changed.

*Referee Comment:*

Line 137- revise to "Microcosm experiments were performed in experimental triplicate…."

*Author Response:*

Accepted and changed.

*Referee Comment:*

Line 141-142; revise "were" to "was"; rainwater is singular.

*Author Response:*

Accepted and changed.

*Referee Comment:*

Line 146; I presume you mean "remove any remaining air bubbles…"

*Author Response:*

Yes, this is correct. Accepted and changed.

*Referee Comment:*

Lines 192-195; consider revising to "At each sampling time, sample splits were preserved without further filtration (<10um) and filtered at 0.02 µm (add filter details). Additionally, at 2,5 and 9 days an additional sample split was filtered at 0.45 µm for colloid characterization." What was the filter type for the 0.45 um filter?

*Author Response:*

We used a Polytetrafluoroethylene Hydrophilic syringe filters (BGB, Boeckten, Switzerland) 0.45µm filters. We added details about 0.45µm filters in this section.

*Referee Comment:*

Line 102 – DOC should be reported in units of mg/L, for consistency with incubation results.

*Author Response:*

Accepted and changed.

*Referee Comment:*

Line 206 – revise to "filtered" fractions. And, it is not common to use "suffix" to describe a subscript, which is what is presented for each term.

*Author Response:*

Thank you for this suggestion. We revised to "filtered" fractions and replaced "suffix" with "subscript".

*Referee Comment:*

Line 231 – revise to "0.5% HCl and 1.0% HNO3".

*Author Response:*

Changed to "1.0 % HNO3 and 0.5 % HCl"

*Referee Comment:*

Line 249 and 275 – consider revision to "NO3- depletion" or "exhausted".

*Author Response:*

Accepted. We changed "reduction" to "depletion".

*Referee Comment:*

Line 260-262; this sentence doesn't make sense and needs revision.

*Author Response:*

This sentence was revised in the scope of revising the results section and figures. We drew new figures including $NO_3^-$, Mn, Fe and $[SO_4^{2-}]$:$[Cl^-]$ together with DOC and discussed soil reduction in detail.

*Referee Comment:*

Line 261 – when describing concentrations in the text, the same units should be used as presented in the figure. Figure 2 presents Mn in units of mg/L.

*Author Response:*

Accepted and changed.

*Referee Comment:*

Line 270 – The reader probably won't remember the "cornfield soil" is the HMLC soils. See my comment above to just spell out the soil type. Consider revising "throughout the experiment" to "over both flooding periods".

*Author Response:*

We acknowledge and share the referees view, that abbreviations are not an optimal way to discuss the data. Spelling out the soil types would be confusing, as the two soil samples only differ in land use and not in soil type. As is was not the goal of this study to compare land use types. We arrived to the conclusion that naming the soils after their mercury content and organic carbon content would be a more approachable practice. Thus, we keep the abbreviations as they are now.

*Referee Comment:*

Figure 4 caption; it is entirely unclear what is meant by "Details on the deconvolution procedure are provided in the supplement".

*Author Response:*

Accepted. Added: "The fractograms of all analysed time points are shown in the supplement" (Figs. S9-S12).

*Referee Comment:*

Figure 4 – should the y-axis label for the top panel indicate "particulate" and should state "total Hg".

*Author Response:*

Accepted and added. $HgT_{<0.45\mu m}$. We did not add "particulate". In most soil science studies samples filtered at 0.45μm are considered "dissolved". Instead of confusing the reader with various terms (...0.45μm = dissolved ; 0.02μm = truly dissolved ;<1kDa =...),  we included the cut off sizes as a subscript.

*Referee Comment:*

Section 3.2 – Consider finding locations in this section to emphasis that you're looking at time points across the two flooding events. Visually, the size proportion of Hg species data look interesting as they show trends in the first flooding period and little change after that.

*Author Response:*

As you suggested we assigned phases and used them to indicate that we are looking the evolution of colloidal Hg through time in the discussion.

*Referee Comment:*

Section 3.3; the sub-header title should specify this is for the 'soil'.

*Author Response:*

Accepted and changed.

*Referee Comment:*

Line 329-330; I don't agree with this conclusion regarding the association of Hg to particulate Mn. In looking at Figure 2, the relative proportion of particulate Mn and Hg decreases with flooding time, but their overall concentration is still low. It is more likely that Hg is release from the soil. The decomposition (and solubilization) of OC in the soils can also release Hg. The pore water DOC concentration is reflecting both release and utilization of DOC, so may not necessarily capture the role of DOC on the Hg mobilization due to carbon mineralisation.

*Author Response:*

Thank you for this input. We can not conclusively say that this is the only process responsible for the increase in dissolved Hg. But especially during Mn reduction the oxidation of NOM compounds can be pronounced. We added references giving evidence of enhanced NOM degradation during Mn oxide reduction and discussed this as an alternative process. Mn reductive dissolution was driving both the release of Hg sorbed to Mn-hydroxides and the degradation and mineralisation of Hg-NOM.

*Referee Comment:*

Lines 346-349 – can you discount the possibility that soil heterogeneity could have contributed to the observed variability?

*Author Response:*

We are convinced that the homogenisation procedure used (Section 2.1) and amount of fresh soil used for an incubation (ca. 500g) made a soil heterogeneity effect unlikely.

*Referee Comment:*

Lines 367-368 – the pore water data strongly support that sulphate reduction is occurring, which show drastic decreases in the concentration of sulphate with increased flooding time. In microcosms of this nature, several biogeochemical processes are occurring simultaneously and the Eh of the system isn't sufficient to assess if sulphate reduction is or is not occurring. My assessment is that it is more likely that sulfate reduction resulted in the re-association of porewater Hg with soils, compared to the NOM complexation.

*Author Response:*

Sulphate concentrations were not directly indicative of the onset of sulphate reduction. This is due to a chemical gradient between supernatant rain water and soils solution demonstrated by the continuous decrease in concentration of conservative ions ($Cl^-$, $Na^+$, $K^+$) (Sect. 4.4). To monitor sulphate reduction, we therefore now added the molar ratios of $SO_4^{2-}$ to $Cl^-$. During sulphate reduction, sulphate to chloride ratios will decrease as in geochemical terms Cl- is a conservative species.

We agree that the Eh measured through a suction cup is not a sufficient measure in complex soil system due to the range of redox presents simultaneously in different pore sizes. We are convinced that sulphate reduction takes place within the incubators. And that Hg is scavenged or reassociated by the formation of ß-HgS in soils. However, one cannot conclusively tell if either this or the complexation NOM is the governing process, wee therefore discuss both hypothesis in the discussion.

*Referee Comment:*

Line 378 – Poulin et al 2016 shows distinct Hg(0) formation in contaminated soil incubations, and should be cited here.

*Author Response:*

Accepted and Cited.

*Referee Comment:*

Lines 380 – one would need citations for the sentence on abiotic vs biotic reduction.

*Author Response:*

We revised this paragraph and added citations for both biotic and abiotic reduction in the dark. UV-light reduction does not apply to our experiments, as they were conducted in the dark.

*Referee Comment:*

Lines 436-438 – one interpretation is that the soils had been subject to period soil flooding that contributed to mercury methylation.

*Author Response:*

This is a valuable point made. We did not monitor the flooding history prior to the sampling. Therefore, we discussed the possibility of a flooding event which induced MeHg levels prior to the field sampling.

*Referee Comment:*

Lines 444-445; could the higher microbial activity be the result of addition of labile carbon? The author should consider highlighting here the diversity of microbial communities that can methylate Hg, as is provided in the Introduction. Sulphate reducing bacteria, metal reducing bacteria, and fermenters are possible contributors to mercury methylation.

*Author Response:*

A higher microbial activity could indeed be the result of addition of labile carbon in the form of manure. We revised this paragraph and highlighted the diversity of Hg methylators using the references given in the introduction.

*Referee Comment:*

Figure 2 – the y-axis labels are very hard to read, and would be nearly impossible to read in print form. Consider re-working the figures as I suggest above, where all terminal electron acceptor processes and DOC are included in a single figure, then all Hg measurements, then all other trace metals.

*Author Response:*

We understand and support the input, created new figures and increased the figure sizes. In the new version of the manuscript three separate figures are presented per soil, highlighting redox processes, Hg measurements, trace elements. They are attached at the end of this document.

We excluded a selection of measured trace metals in the article as the discussion of all the measured trace metals is beyond the scope of this manuscript.

***Referee Comment:***

SI Specific Comments:

Line 7 – this figure should be revised to state "total Hg" when total Hg is measured. This needs to be fixed in all cases in text and figures, in the main text and SI.

***Author Response:***

Accepted and changed for all the figures and in the text.

[Figure]

*Figure 1 Soil solution dynamics in cornfield soil (HMLC) incubations for redox potential (a), redox reactive elements (Mn, PMn, Fe, P-Fe, $[SO_4^{2-}]:[Cl^-]$) (b-f) and dissolved organic carbon (h). Lines between points were plotted to improve readability. The gray area indicates the drained period.*

[Figure]

*Figure 2 Soil solution dynamics in cornfield soil (HMLC) incubations for Hg (a-c) subdivided in phases (0-3). Lines between points were plotted to improve readability. The gray area indicates the drained period. Red arrows indicate sampling days for AF4-ICP-MS analyses.*

[Figure]

*Figure 3 Soil solution dynamics in cornfield soil (HMLC) incubations for Cu (a) and As (b). Lines between points were plotted to improve readability. The gray area indicates the drained period.*

[Figure]

*Figure 4 Soil solution dynamics in pasture field soil (LMHC) incubations for redox potential (a), redox reactive elements (Mn, PMn, Fe, P-Fe, [SO₄²⁻]:[Cl⁻]) (b-f) and dissolved organic carbon (h). Lines between points were plotted to improve readability. The gray area indicates the drained period.*

[Figure]

*Figure 5 Soil solution dynamics in pasture field soil (LMHC) incubations for Hg (a-c) subdivided in phases (1-3). Lines between points were plotted to improve readability. The gray area indicates the drained period.*

[Figure]

*Figure 6 Soil solution dynamics in pasture field soil (LMHC) incubations for Cu (a) and As (b). Lines between points were plotted to improve readability. The gray area indicates the drained period.*